# Prototypical Graph ODE for Modeling Complicated Interacting Dynamics

## Abstract

This paper studies the problem of modeling multi-agent dynamical systems, where agents could interact mutually to influence their behaviors. Recent research predominantly uses geometric graphs to depict these mutual interactions, which are then captured by powerful graph neural networks (GNNs). However, predicting interacting dynamics in challenging scenarios such as out-of-distribution shift and complicated underlying rules remains unsolved. In this paper, we propose a new approach named Prototypical Graph ODE (PGODE) to address the problem. The core of PGODE is to incorporate prototype decomposition from contextual knowledge into a *continuous* graph ODE framework. Specifically, PGODE employs representation disentanglement and system parameters to extract both object-level and system-level contexts from historical trajectories, which allows us to explicitly model their independent influence and thus enhances the *generalization* capability under system changes. Then, we integrate these disentangled latent representations into a graph ODE model, which determines a combination of various interacting prototypes for enhanced model *expressivity*. The entire model is optimized using an end-to-end variational inference framework to maximize the likelihood. Extensive experiments in both in-distribution and out-of-distribution settings validate the superiority of PGODE.

## 1 Introduction

Multi-agent dynamical systems are ubiquitous in the real world where agents can be vehicles (Yıldız et al., 2022) and microcosmic particles (Shao et al., 2022). These agents could have complicated interactions resulting from behavioral or mechanical influences, which result in complicated future trajectories of the whole system. Modeling the interacting dynamics is a crucial challenge in machine learning with broad applications in fluid mechanics (Pfaff et al., 2021; Mayr et al., 2023), autonomous driving (Yu et al., 2020; Zhu et al., 2023) and human-robot (Schaefer et al., 2021; Abeyruwan et al., 2023). Extensive time-series approaches based on recurrent neural networks (Weerakody et al., 2021) and Transformers (Zhou et al., 2021) are generally designed for single-agent dynamical systems (Fotiadis et al., 2023), which fall short when it comes to capturing the intricate relationships among interacting objects. To address this gap, geometric graphs (Kofinas et al., 2021) are usually employed to represent the interactions between objects where nodes represent individual objects, and edges are built when a connection exists between two nodes. These connections can be obtained from geographical distances between atoms in molecular dynamics (Li et al., 2022b) and underlying equations in mechanical systems (Huang et al., 2020).

In the literature, graph neural networks (GNNs) (Kipf & Welling, 2017; Xu et al., 2019a; Zheng et al., 2022; Li et al., 2022a; He et al., 2022) have been increasingly prevailing for learning from geometric graphs in interacting dynamical systems (Pfaff et al., 2021; Shao et al., 2022; Sanchez-Gonzalez et al., 2020; Han et al., 2022; Meirom et al., 2021; Yıldız et al., 2022). These GNN-based approaches primarily focus on predicting the future states of dynamic systems with the message passing mechanism. Specifically, they begin with encoding the states of trajectories and then iteratively update each node representation by incorporating information from its adjacent nodes, which effectively captures the complex interacting dynamics among the objects in systems.

Despite the significant advancements, GNN-based approaches often suffer from performance decreasement in challenging scenarios such as long-term dynamics (Lippe et al., 2023), complicated

governing rules (Gu et al., 2022), and out-of-distribution shift (Dendorfer et al., 2021). As a consequence, developing a high-quality data-driven model requires us to consider the following critical points: (1) *Capturing Continuous Dynamics*. The majority of existing methods predict the whole trajectories in an autoregressive manner (Pfaff et al., 2021; Shao et al., 2022; Sanchez-Gonzalez et al., 2020), which iteratively feed next-time predictions back into the input. These rollouts could lead to error accumulation and thus fail to capture long-term dynamics accurately. (2) *Expressivity*. There are a variety of interacting dynamical systems governed by complex partial differential equations (PDEs) in physics and biology (Rao et al., 2023). Therefore, a high-quality model with strong *expressivity* is anticipated for sufficient learning. (3) *Generalization*. In practical applications, the distributions of training and test trajectories could differ due to variations in system parameters (Sanchez-Gonzalez et al., 2020; Li et al., 2023). Current data-driven models could perform poorly when confronting system changes during the inference phase (Goyal & Bengio, 2022).

In this paper, we propose a novel approach named Prototypical Graph ODE (PGODE) for complicated interacting dynamics modeling. The core of PGODE lies in exploring disentangled contexts, i.e., object states and system states, inferred from historical trajectories for graph ODE with high *expressivity* and *generalization*. To begin, we extract both object-level and system-level contexts via message passing and attention mechanisms for subsequent dynamics modeling. Object-level contexts refer to individual attributes such as initial states and local heterophily (Luan et al., 2022), while system-level contexts refer to shared parameters such as temperature and viscosity. To improve generalization under system changes, we focus on two strategies. First, we enhance the invariance of object-level contexts under system changes through representation disentanglement. Second, we establish a connection between known system parameters and system-level latent representations. Furthermore, we incorporate this contextual information into a graph ODE framework to capture long-term dynamics through *continuous* evolution instead of discrete rollouts. More importantly, we introduce a set of learnable GNN prototypes that can be trained to represent different interaction patterns. The weights for each object are then derived from its hierarchical representations to provide individualized dynamics. Our framework can be illustrated from a mixture-of-experts perspective, which boosts the *expressivity* of the model. Finally, we integrate our method into an end-to-end variational inference framework to optimize the evidence lower bound (ELBO) of the likelihood. Comprehensive experiments in different settings validate the superiority of PGODE.

The contributions of this paper can be summarized in three points: (1) *New Connection*. To the best of our knowledge, this work is the first to connect context mining with a prototypical graph ODE approach for modeling challenging interacting dynamics. (2) *Methodology*. We extract hierarchical contexts with representation disentanglement and system parameters, which are then integrated into a graph ODE model that utilizes prototype decomposition. (3) *Superior Performance*. Extensive experiments validate the efficacy of our approach in different challenging settings.

## 2 BACKGROUND

**Problem Definition.** Given a multi-agent dynamical system, we characterize the agent states and interaction at the $t$-th timestamp as a graph $G^t = (\mathcal{V}, \mathcal{E}^t, \boldsymbol{X}^t)$, where each node in $\mathcal{V}$ is an object, $\mathcal{E}^t$ comprises all the edges and $\boldsymbol{X}^t$ is the object attribute matrix. $N$ represents the number of objects. Given the observations $G^{1:T_{obs}} = \{G^1, \cdots, G^{T_{obs}}\}$, our goal is to learn a model capable of predicting the future trajectories $\boldsymbol{X}^{T_{obs}+1:T}$. Our interacting dynamics system is governed by a set of equations with system parameters denoted as $\boldsymbol{\xi}$. Different values of $\boldsymbol{\xi}$ could influence underlying dynamical principles, leading to potential shift in trajectory distributions. Therefore, it is essential to extract contextual information related to both system parameters and node states from historical observations for faithful trajectory predictions.

**Neural ODEs for Multi-agent Dynamical Systems.** Neural ODEs have been shown effective in modeling various dynamical systems (Chen et al., 2018; Huang et al., 2021). For single-agent dynamical systems, the evolution of latent representations $\boldsymbol{z}^t$ can be expressed via a given ODE $\frac{d\boldsymbol{z}^t}{dt} = f(\boldsymbol{z}^t)$. Then, the entire trajectory of the system can be determined using $\boldsymbol{z}^T = \boldsymbol{z}^0 + \int_{t=0}^{T} f(\boldsymbol{z}^t) \, dt$. For multi-agent dynamical systems, the formulation can be extended as follows:

$$\boldsymbol{z}_i^T = \boldsymbol{z}_i^0 + \int_{t=0}^{T} f_i\left(\boldsymbol{z}_1^t, \boldsymbol{z}_2^t \cdots \boldsymbol{z}_N^t\right) dt, \tag{1}$$

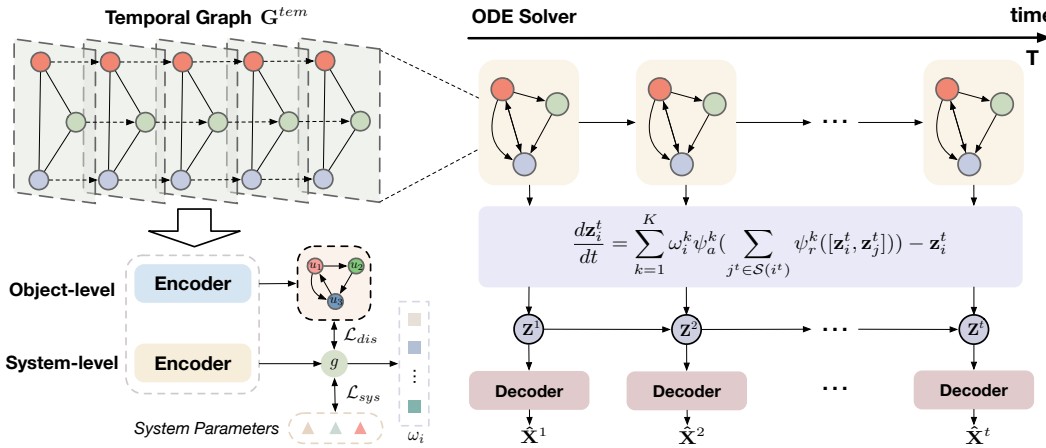

Figure 1: An overview of the proposed PGODE. PGODE first constructs a temporal graph and then utilizes different encoders to extract object-level and system-level contexts using representation disentanglement and system parameters. These contexts would generate weights for a prototypical graph ODE framework, which models the evolution of objects. Finally, object representations are fed into a decoder to output the states at any timestamp.

where $\boldsymbol{z}_i^t$ represents the latent state embedding for object $i$ at timestamp $t$, and $f_i$ models the interacting dynamics specifically for object $i$. With Eqn. 1, we can calculate $\boldsymbol{z}_i^t$ using numerical solvers such as Runge-Kutta (Schober et al., 2019) and Leapfrog (Zhuang et al., 2021), which produce accurate predictions of future trajectories in the multi-agent systems.

## 3 THE PROPOSED APPROACH

This paper introduces a novel approach PGODE for modeling interacting system dynamics in challenging scenarios such as out-of-distribution shift and complicated underlying rules. The core of PGODE lies in exploring disentangled contexts for prototype decomposition for a high-quality graph ODE framework. Specifically, we first construct a temporal graph to learn disentangled object-level and system-level contexts from historical data and system parameters. These contexts further determine prototype decomposition, which characterizes distinct interacting patterns in a graph ODE framework for modeling continuous dynamics. We adopt a decoder to output the trajectories and the whole model is optimized via an end-to-end variational inference framework. An overview of PGODE is depicted in Figure 1, and the details will be presented below.

### 3.1 HIERARCHICAL CONTEXT DISCOVERY WITH DISENTANGLEMENT

A promising solution to formulating the dynamics of interacting systems is the introduction of GNNs into Eqn. 1 where different GNNs are tailored for distinct nodes across diverse systems. Generally, the interacting dynamics of each object are influenced by both system-level and object-level contexts. System-level contexts include temperature, viscosity, and coefficients in underlying equations (Rämä & Sipilä, 2017), which are shared in the whole system. Object-level contexts refer to object attributes such as initial states, and local heterophily (Luan et al., 2022), which give rise to distinct interacting patterns for individual objects. To design GNNs for a variety of objects and system configurations, it is essential to derive object-level and system-level latent embeddings from historical trajectories. Additionally, note that system parameters could differ between training and test datasets (Kim et al., 2021), thereby leading to potential distribution shift. To mitigate its influence, we disentangle object-level and system-level embeddings with known system parameters for a more precise and independent description of complex dynamical systems.

**Object-level Contexts.** We aim to condense the historical trajectories into informative object representations. To achieve this, we conduct the message passing procedure on a temporal graph for

observation representation updating. Then, object representations are generated by summarizing all the observations using the attention mechanism (Niu et al., 2021).

In detail, a temporal graph is first constructed where each node represents an observation, and edges represent temporal and spatial relationships. Temporal edges connect consecutive observations of the same object, while spatial edges are built when observations from two different objects are connected at the same timestamp. In formulation, we have $NT^{obs}$ nodes in the temporal graph $G^{tem}$ and its adjacency matrix can be written as:

$$\boldsymbol{A}^{tem}(i^t, j^{t'}) = \begin{cases} w_{ij}^t & t = t', , \\ 1 & i = j, t' = t + 1, \\ 0 & \text{otherwise,} \end{cases} \tag{2}$$

where $i^t$ represents the observation of $i$ at timestamp $t$ and $w_{ij}^t$ is the edge weight from $G^t$. Then, we adopt the message passing mechanism to learn from the temporal graph. Denote the representation of $i^t$ at the $l$-th layer as $\boldsymbol{h}_i^{t,(l)}$, and the interaction scores can be obtained by comparing representations between the query and key spaces as follows:

$$\alpha^{(l)}(i^s, j^{s'}) = \frac{1}{\sqrt{d}} \boldsymbol{A}^{tem}(i^s, j^{s'})(\boldsymbol{W}_{query}\hat{\boldsymbol{h}}_i^{s,(l)})^T(\boldsymbol{W}_{key}\hat{\boldsymbol{h}}_j^{s',(l)}), \tag{3}$$

where $d$ denotes the hidden dimension and $\hat{\boldsymbol{h}}_i^{t,(l)} = \boldsymbol{h}_i^{t,(l)} + TE(t)$. Here $TE(t)$ is the temporal embedding with $\text{TE}(t)[2i] = \sin\left(\frac{t}{10000^{2i/d}}\right)$ and $\text{TE}(t)[2i+1] = \cos\left(\frac{t}{10000^{2i/d}}\right)$. $\boldsymbol{W}_{query} \in \mathbb{R}^{d \times d}$ and $\boldsymbol{W}_{key} \in \mathbb{R}^{d \times d}$ are two weight matrices for feature transformation. Then, we update each representation by aggregation semantics from its neighborhood as follows:

$$\boldsymbol{h}_i^{s,(l+1)} = \boldsymbol{h}_i^{s,(l)} + \sigma\left(\sum_{j^{s'} \in \mathcal{S}(i^s)} \alpha^{(l)}(i^s, j^{s'})\boldsymbol{W}_{value}\hat{\boldsymbol{h}}_j^{s',(l)}\right), \tag{4}$$

where $\boldsymbol{W}_{value} \in \mathbb{R}^{d \times d}$ is to project representations into values and $\mathcal{S}(\cdot)$ collects all the neighboring nodes. Finally, we summarize all observation representations for every object $i$ into a latent representation $\boldsymbol{u}_i$ using the attention mechanism:

$$\boldsymbol{q}_i^t = \boldsymbol{h}_i^{t,(L)} + \text{TE}(t), \quad \boldsymbol{u}_i = \frac{1}{N^{obs}} \sum_{t=1}^{N^{obs}} \sigma(\boldsymbol{W}_{sum}\boldsymbol{q}_i^t), \tag{5}$$

in which $\boldsymbol{W}_{sum}$ is for feature transformation. In this manner, we incorporate semantics from both the observed trajectories and geometric structures into expressive object-level latent representations, i.e., $\{\boldsymbol{u}_i\}_{i=1}^N$ for predicting future complicated interacting dynamics in systems.

**System-level Contexts.** In real-world applications, system parameters may vary between training and test datasets, leading to out-of-distribution shift in trajectories (Mirza et al., 2022; Ragab et al., 2023). To robustly capture these variations and enhance model performance, we employ a separate network to infer system-level contexts from historical trajectories, which are guided by system parameters in the training data. Moreover, we employ mutual information minimization to disentangle object-level and system-level representations, which allows for a clear separation of influences and thus enables the invariance of object-level contexts under system changes.

In particular, we employ the same network architecture but with different parameters to generate the latent representation $\boldsymbol{u}_i'$ for object $i$. Then, a pooling operator is adopted to summarize all these object-level representations into a system-level representation $\boldsymbol{g}$ as $\boldsymbol{g} = \sum_{i=1}^N \boldsymbol{u}_i'$. To capture contexts from system parameters, we maximize the mutual information between the system-level representation and known parameters, i.e., $I(\boldsymbol{g}; \boldsymbol{\xi})$. Meanwhile, to disentangle object-level and system-level latent representation, we minimize their mutual information, i.e., $I(\boldsymbol{g}; \boldsymbol{u}_i)$, which enables us to better handle the variations introduced by out-of-distribution system parameters. In our implementation, we make use of Jensen-Shannon mutual information estimator $T_\gamma(\cdot, \cdot)$ (Chen et al., 2019) with parameters $\gamma$, and the loss objective for learning system parameters can be:

$$\mathcal{L}_{sys} = \frac{1}{|\mathcal{P}|} \sum_{(\boldsymbol{g},\boldsymbol{\xi}) \in \mathcal{P}} -sp(-T_\gamma(\boldsymbol{g}, \boldsymbol{\xi})) + \frac{1}{|\mathcal{P}|^2} \sum_{(\boldsymbol{g},\boldsymbol{\xi}) \notin \mathcal{P}} sp(-T_\gamma(\boldsymbol{g}, \boldsymbol{\xi})), \tag{6}$$

where $sp(\boldsymbol{x}) = \log(1 + e^{\boldsymbol{x}})$ denotes the softplus function and $\mathcal{P}$ collects all the positive pairs from the same system. Similarly, the loss objective for representation disentanglement is formulated as:

$$\mathcal{L}_{dis} = max_{\gamma'}\{\frac{1}{|\mathcal{P}'|}\sum_{(\boldsymbol{g},\boldsymbol{u}_i)\in\mathcal{P}'} sp(-T_{\gamma'}(\boldsymbol{g},\boldsymbol{u}_i)) + \frac{1}{|\mathcal{P}'||\mathcal{P}|}\sum_{(\boldsymbol{g},\boldsymbol{u}_i)\notin\mathcal{P}'} -sp(-T_{\gamma'}(\boldsymbol{g},\boldsymbol{u}_i))\}, \quad (7)$$

where $T_{\gamma'}$ is optimization in an adversarial manner and $\mathcal{P}'$ collects all the positive object-system pairs. Differently, $T_{\gamma'}$ is trained adversarially for precise measurement of mutual information. On this basis, we establish the connection between system-level contexts and explicit parameters while simultaneously minimizing their impact on the object-level contexts through representation disentanglement. In this way, our model separates and accurately captures the influence of these two factors, enhancing the generalization capacity when system parameters vary during evaluation.

## 3.2 PROTOTYPICAL GRAPH ODE

After extracting context embeddings, we intend to integrate them into a graph ODE framework for multi-agent dynamic systems. However, training a separate GNN for each node would introduce an excessive number of parameters, which could result in overfitting and a complicated optimization process (Zhao et al., 2020). To address this, we learn a set of GNN prototypes to characterize the entire GNN space, and then use prototype decomposition for each object in the graph ODE. Specifically, we start by initializing state representations for each node and then determine the weights for each object based on both object-level and system-level contexts.

To begin, we utilize object-level contexts with feature transformation for initialization. Here, the initial state representations are sampled from an approximate posterior distribution $q(\boldsymbol{z}_i^0|G^{tem})$, which would be close to a prior distribution $p(\boldsymbol{z}_i^0)$. The mean and variance are learned from $\boldsymbol{u}_i$ as:

$$q\left(\boldsymbol{z}_i^0 \mid G^{tem}\right) = \mathcal{N}\left(\psi^m\left(\boldsymbol{u}_i\right), \psi^v\left(\boldsymbol{u}_i\right)\right), \quad (8)$$

where $\psi^m(\cdot)$ and $\psi^v(\cdot)$ are two feed-forward networks (FFNs) to compute the mean and variance. Then, we introduce $K$ GNN prototypes, each with two FFNs $\psi_r^k(\cdot)$ and $\psi_a^k(\cdot)$ for relation learning and feature aggregation, respectively. The updating rule of the $k$-th prototypes for object $i$ is formulated as follows:

$$f_i^k\left(\boldsymbol{z}_1^t, \boldsymbol{z}_2^t \cdots \boldsymbol{z}_N^t\right) = \psi_a^k(\sum_{j^t\in\mathcal{S}(i^t)} \psi_r^k([\boldsymbol{z}_i^t, \boldsymbol{z}_j^t])), \quad (9)$$

where $j^t$ represents the neighbor of $i$ at timestamp $t$. Then, we take a weighted combination of these GNN prototypes for each object, and the prototypical interacting dynamics can be formulated as:

$$\frac{d\boldsymbol{z}_i^t}{dt} = \sum_{k=1}^{K}\boldsymbol{w}_i^k\psi_a^k(\sum_{j^t\in\mathcal{S}(i^t)} \psi_r^k([\boldsymbol{z}_i^t, \boldsymbol{z}_j^t])) - \boldsymbol{z}_i^t. \quad (10)$$

The last term indicates natural recovery, which usually benefits semantics learning in practice. To generate the weights for each object, we merge both object-level and system-level latent variables and adopt a FFN $\rho(\cdot)$ as follows:

$$\boldsymbol{w}_i = [\boldsymbol{w}_i^1, \cdots, \boldsymbol{w}_i^K] = \rho([\boldsymbol{u}_i, \boldsymbol{g}]), \quad (11)$$

where the softmax activation is adopted to ensure $\sum_{k=1}^{K}\boldsymbol{w}_i^k = 1$.

**A Mixture-of-Experts Perspective.** We will demonstrate that our graph ODE model can be interpreted through the lens of the mixture of experts (MoE) (Du et al., 2022). Specifically, each prototype serves as an ODE expert, while $\boldsymbol{w}_i$ acts as the gating weights that control the contribution of each expert. Through this, we are the first to get the graph ODE married with MoE, enhancing the expressivity to capture complex interacting dynamics. More importantly, different from previous works that employ black-box routing functions (Zhou et al., 2022), the routing function in our PGODE is derived from hierarchical contexts with representation disentanglement, which further equips our model with the generalization capability to handle potential shift in data distributions.

**Existence and Uniqueness.** Moreover, we give a theoretical analysis about the existence and uniqueness of our proposed graph ODE to show that it is well-defined under certain conditions.

**Lemma 3.1.** *We first assume that the learnt functions $\psi_r^k : \mathbb{R}^{2d} \to \mathbb{R}^d, \psi_a^k : \mathbb{R}^d \to \mathbb{R}^d$ have bounded gradients. In other words, there exists $A, R > 0$, such that the following Jacobian matrices have the bounded matrix norms:*

$$J_{\psi_r^k}(\boldsymbol{x}, \boldsymbol{y}) = \begin{pmatrix} \frac{\partial \psi_{r,1}^k}{\partial x_1} & \cdots & \frac{\partial \psi_{r,1}^k}{\partial x_d} & \frac{\partial \psi_{r,1}^k}{\partial y_1} & \cdots & \frac{\partial \psi_{r,1}^k}{\partial y_d} \\ \vdots & \ddots & \vdots & \vdots & \ddots & \vdots \\ \frac{\partial \psi_{r,d}^k}{\partial x_1} & \cdots & \frac{\partial \psi_{r,d}^k}{\partial x_d} & \frac{\partial \psi_{r,d}^k}{\partial y_1} & \cdots & \frac{\partial \psi_{r,d}^k}{\partial y_d} \end{pmatrix}, \quad \|J_{\psi_r^k}(\boldsymbol{x}, \boldsymbol{y})\| \le R, \quad (12)$$

$$J_{\psi_a^k}(\boldsymbol{x}) = \begin{pmatrix} \frac{\partial \psi_{a,1}^k}{\partial x_1} & \cdots & \frac{\partial \psi_{a,1}^k}{\partial x_d}, \\ \vdots & \ddots & \vdots \\ \frac{\partial \psi_{a,d}^k}{\partial x_1} & \cdots & \frac{\partial \psi_{a,d}^k}{\partial x_d} \end{pmatrix}, \quad \|J_{\psi_a^k}(\boldsymbol{x})\| \le A. \quad (13)$$

*Then, given the initial state $(t_0, \boldsymbol{z}_1^{t_0}, \cdots, \boldsymbol{z}_N^{t_0}, \boldsymbol{w}_1, \cdots, \boldsymbol{w}_N)$, we claim that there exists $\varepsilon > 0$, such that the ODE system Eqn. 10 has a unique solution in the interval $[t_0 - \varepsilon, t_0 + \varepsilon]$.*

The proof is shown in Appendix A. Our analysis demonstrates that based on given observations, future trajectories are predictable using our graph ODE, which is an essential property in interacting dynamics modeling (Chen et al., 2018; Kong et al., 2020).

### 3.3 DECODER AND OPTIMIZATION

Finally, we introduce a decoder to forecast future trajectories, along with an end-to-end variational inference framework for the maximization of the likelihood.

In particular, we build a connection between latent states and trajectories by calculating the likelihood for each observation $p(\boldsymbol{x}_i^t | \boldsymbol{z}_i^t)$. Following the maximum likelihood estimation of a Gaussian distribution, here we solely produce the mean of each distribution, i.e., $\boldsymbol{\mu}_i^t = \phi(\boldsymbol{z}_i^t)$, where $\phi(\cdot)$ is an FFN serving as the decoder implemented. In the variational inference framework, we maximize the evidence lower bound (ELBO) of the likelihood, which involves the maximization of the likelihood for observed trajectories and the minimization of the divergence between the prior and posterior distributions. Formally,

$$\mathcal{L}_{elbo} = \mathbb{E}_{Z^0 \sim \prod_{i=1}^N q(\boldsymbol{z}_i^0 | G^{1:T_{obs}})} \left[ \log p(\boldsymbol{X}^{T_{obs}+1:T}) \right] - \text{KL} \left[ \prod_{i=1}^N q(\boldsymbol{z}_i^0 | G^{1:T_{obs}}) \| p\left(\boldsymbol{Z}^0\right) \right], \quad (14)$$

in which $KL(\cdot\|\cdot)$ outputs the Kullback-Leibler (KL) divergence. Eqn. 14 can be re-written into the following equation by incorporating the independence of each node:

$$\mathcal{L}_{elbo} = - \sum_{i=1}^N \sum_{t=T_{obs}+1}^T \frac{\|\boldsymbol{x}_i^t - \boldsymbol{\mu}_i^t\|^2}{2\sigma^2} - \text{KL} \left[ \prod_{i=1}^N q(\boldsymbol{z}_i^0 | G^{1:T_{obs}}) \| p\left(\boldsymbol{Z}^0\right) \right], \quad (15)$$

in which $\sigma^2$ represents the variance of the prior distribution. To summarize, the final loss objective for the optimization is written as:

$$\mathcal{L} = \mathcal{L}_{elbo} + \mathcal{L}_{sys} + \mathcal{L}_{dis}, \quad (16)$$

where the last two loss terms serve as a regularization mechanism using mutual information to constrain the model parameters (Xu et al., 2019b; Rhodes & Lee, 2021). We have summarized the whole algorithm in Appendix D.

## 4 EXPERIMENT

Our proposed GOAT is evaluated on both physical and molecular dynamical systems. Each trajectory sample is further split into two parts, i.e., a conditional part for initializing object-level context representations and global-level context representations, and a prediction part for supervision. We denote the size of the two parts as conditional length and prediction length, respectively. Our approach is compared with seven baselines, i.e., LSTM (Hochreiter & Schmidhuber, 1997), GRU (Cho et al., 2014), NODE (Chen et al., 2018), LG-ODE (Huang et al., 2020), MPNODE (Chen et al., 2022), SocialODE (Wen et al., 2022) and HOPE (Luo et al., 2023). The details about in-distribution (ID) and out-of-distribution (OOD) settings are in Appendix G.

Table 1: Mean Squared Error (MSE) $\times 10^{-2}$ on physical dynamics simulations.

| Prediction Length | 12 (ID) | | 24 (ID) | | 36 (ID) | | 12 (OOD) | | 24 (OOD) | | 36 (OOD) | |
|---|---|---|---|---|---|---|---|---|---|---|---|---|
| Variable | $q$ | $v$ | $q$ | $v$ | $q$ | $v$ | $q$ | $v$ | $q$ | $v$ | $q$ | $v$ |
| *Springs* | | | | | | | | | | | | |
| LSTM | 0.287 | 0.920 | 0.659 | 2.659 | 1.279 | 5.729 | 0.474 | 1.157 | 0.938 | 2.656 | 1.591 | 5.223 |
| GRU | 0.394 | 0.597 | 0.748 | 1.856 | 1.248 | 3.446 | 0.591 | 0.708 | 1.093 | 1.945 | 1.671 | 3.423 |
| NODE | 0.157 | 0.564 | 0.672 | 2.414 | 1.608 | 6.232 | 0.228 | 0.791 | 0.782 | 2.530 | 1.832 | 6.009 |
| LG-ODE | 0.077 | 0.268 | 0.155 | 0.513 | 0.527 | 2.143 | 0.088 | 0.299 | 0.179 | 0.562 | 0.614 | 2.206 |
| MPNODE | 0.076 | 0.243 | 0.171 | 0.456 | 0.600 | 1.737 | 0.094 | 0.249 | 0.212 | 0.474 | 0.676 | 1.716 |
| SocialODE | 0.069 | 0.260 | 0.129 | 0.510 | 0.415 | 2.187 | 0.079 | 0.285 | 0.153 | 0.570 | 0.491 | 2.310 |
| HOPE | 0.070 | 0.176 | 0.456 | 0.957 | 2.475 | 5.409 | 0.076 | 0.221 | 0.515 | 1.317 | 2.310 | 5.996 |
| PGODE (Ours) | **0.035** | **0.124** | **0.070** | **0.262** | **0.296** | **1.326** | **0.047** | **0.138** | **0.088** | **0.291** | **0.309** | **1.337** |
| *Charged* | | | | | | | | | | | | |
| LSTM | 0.795 | 3.029 | 2.925 | 3.734 | 6.569 | 4.331 | 1.127 | 3.027 | 3.988 | 3.640 | 8.185 | 4.221 |
| GRU | 0.781 | 2.997 | 2.805 | 3.640 | 5.969 | 4.147 | 1.042 | 3.028 | 3.747 | 3.636 | 7.515 | 4.101 |
| NODE | 0.776 | 2.770 | 3.014 | 3.441 | 6.668 | 4.043 | 1.124 | 2.844 | 3.931 | 3.563 | 8.497 | 4.737 |
| LG-ODE | 0.759 | 2.368 | 2.526 | 3.314 | 5.985 | 5.618 | 0.932 | 2.551 | 3.018 | 3.589 | 6.795 | 6.365 |
| MPNODE | 0.740 | 2.455 | 2.458 | 3.664 | 5.625 | 6.259 | 0.994 | 2.555 | 2.898 | 3.835 | 6.084 | 6.797 |
| SocialODE | 0.662 | 2.335 | 2.441 | 3.252 | 6.410 | 4.912 | 0.894 | 2.420 | 2.894 | 3.402 | 6.292 | 6.340 |
| HOPE | 0.614 | 2.316 | 3.076 | 3.381 | 8.567 | 8.458 | 0.878 | 2.475 | 3.685 | 3.430 | 10.953 | 9.120 |
| PGODE (Ours) | **0.578** | **2.196** | **2.037** | **2.648** | **4.804** | **3.551** | **0.802** | **2.135** | **2.584** | **2.663** | **5.703** | **3.703** |

SocialODE      HOPE      GOAT      Ground Truth

Figure 2: Visualization of different methods on *Springs*. Semi-transparent paths denote observed trajectories, from which the latent initial states are estimated. Solid paths denote model predictions.

## 4.1 PERFORMANCE ON PHYSICAL DYNAMICS SIMULATIONS

**Datasets.** We employ two physics simulation datasets to evaluate our proposed GOAT, i.e., *Springs* and *Charged* (Kipf et al., 2018). Each sample in these two simulated datasets contains 10 interacting particles in a 2D box that has no external forces but possible collisions. We aim to predict the future position information and the future velocity values of these interacting particles, i.e., $q$ and $v$. More details of the two datasets can be found in Appendix F.

**Performance Comparison.** The compared results with respect to different prediction lengths are collected in Table 1. From the results, we have two observations. *Firstly,* ODE-based methods generally outperform discrete methods, which validates that continuous methods can naturally capture system dynamics and relieve the influence of potential error accumulation. *Secondly*, our proposed PGODE achieves the best performance among all the methods. In particular, the average MSE reduction of our PGODE over HOPE is 47.40% for ID and 48.57% for OOD settings on these two datasets. The remarkable performance can be attributed to two reasons: (1) Introduction of context discovery. PGODE generates disentangled object-level and system-level embeddings, which would increase the generalization capability of the model to handle system changes, especially in OOD settings. (2) Introduction of prototype decomposition. PGODE combines a set of GNN prototypes to characterize the interacting patterns, which increases the expressivity of the model for complex dynamics. More compared results can be found in Sec. H.1.

**Visualization.** Figure 2 shows the visualization of three compared methods and the ground truth on *Springs*. Here, semi-transparent paths denote the observed trajectories while solid paths denote the predicted ones. From the results, we can observe that our proposed PGODE can generate reliable trajectories close to the ground truth for all the timestamps while both baselines SocialODE and HOPE fail, which validates the superiority of the proposed PGODE.

Table 2: Mean Squared Error (MSE) $\times 10^{-3}$ on molecular dynamics simulations.

| Prediction Length | 12 (ID) | | | 24 (ID) | | | 12 (OOD) | | | 24 (OOD) | | |
|---|---|---|---|---|---|---|---|---|---|---|---|---|
| Variable | $q_x$ | $q_y$ | $q_z$ | $q_x$ | $q_y$ | $q_z$ | $q_x$ | $q_y$ | $q_z$ | $q_x$ | $q_y$ | $q_z$ |
| *5AWL* | | | | | | | | | | | | |
| LSTM | 4.178 | 3.396 | 3.954 | 4.358 | 4.442 | 3.980 | 4.785 | 4.178 | 4.467 | 5.152 | 5.216 | 4.548 |
| GRU | 4.365 | 2.865 | 2.833 | 5.295 | 3.842 | 3.996 | 5.139 | 3.662 | 3.789 | 6.002 | 4.723 | 5.358 |
| NODE | 3.992 | 3.291 | 2.482 | 4.674 | 4.333 | 3.254 | 4.390 | 4.135 | 2.808 | 5.734 | 5.388 | 4.036 |
| LG-ODE | 2.825 | 2.807 | 2.565 | 3.725 | 3.940 | 3.412 | 3.358 | 3.549 | 3.501 | 4.611 | 4.763 | 4.543 |
| MPNODE | 2.631 | 3.029 | 2.734 | 3.587 | 4.151 | 3.488 | 3.061 | 3.899 | 3.355 | 4.271 | 5.085 | 4.427 |
| SocialODE | 2.481 | 2.729 | 2.473 | 3.320 | 3.951 | 3.399 | 2.987 | 3.514 | 3.166 | 4.248 | 4.794 | 4.155 |
| HOPE | 2.326 | 2.572 | 2.442 | 3.495 | 3.816 | 3.413 | 2.581 | 3.528 | 2.955 | 4.548 | 5.047 | 4.007 |
| PGODE (Ours) | **2.098** | **2.344** | **2.099** | **2.910** | **3.384** | **2.904** | **2.217** | **3.109** | **2.593** | **3.374** | **4.334** | **3.615** |
| *2N5C* | | | | | | | | | | | | |
| LSTM | 2.608 | 2.265 | 3.975 | 3.385 | 2.959 | 4.295 | 3.285 | 2.210 | 5.247 | 3.834 | 2.878 | 5.076 |
| GRU | 2.847 | 2.968 | 3.493 | 3.340 | 3.394 | 3.636 | 3.515 | 3.685 | 3.796 | 4.031 | 3.938 | 3.749 |
| NODE | 2.211 | 2.103 | 2.601 | 3.074 | 2.849 | 3.284 | 2.912 | 2.648 | 2.799 | 3.669 | 3.478 | 3.874 |
| LG-ODE | 2.176 | 1.884 | 1.928 | 2.824 | 2.413 | 2.689 | 2.647 | 2.284 | 2.326 | 3.659 | 3.120 | 3.403 |
| MPNODE | 1.855 | 1.923 | 2.235 | 2.836 | 2.805 | 3.416 | 2.305 | 2.552 | 2.373 | 3.244 | 3.537 | 3.220 |
| SocialODE | 1.965 | 1.717 | 1.817 | 2.575 | 2.286 | 2.412 | 2.348 | 2.138 | 2.169 | 3.380 | 2.990 | 3.057 |
| HOPE | 1.842 | 1.915 | 2.223 | 2.656 | 2.788 | 3.474 | 2.562 | 2.514 | 2.731 | 3.343 | 3.301 | 3.502 |
| PGODE (Ours) | **1.484** | **1.424** | **1.575** | **1.960** | **2.029** | **2.119** | **1.684** | **1.809** | **1.912** | **2.464** | **2.734** | **2.727** |

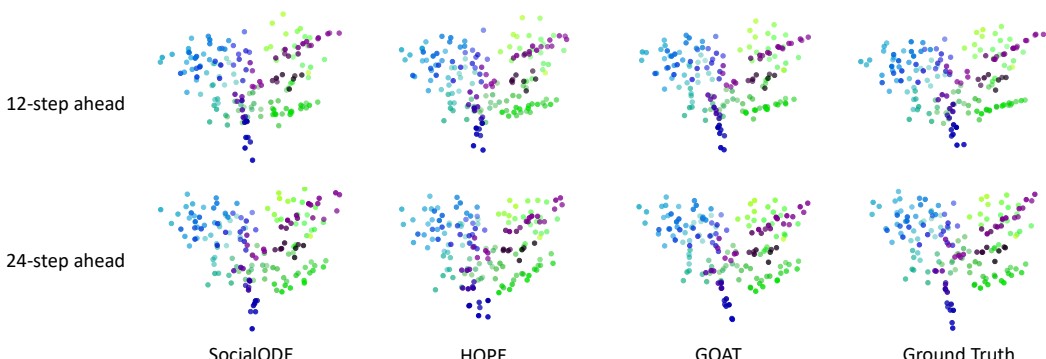

Figure 3: Visualization of prediction results of different methods on the *5AWL* dataset. We can observe that our PGODE can reconstruct the ground truth accurately.

## 4.2 PERFORMANCE ON MOLECULAR DYNAMICS SIMULATIONS

**Datasets.** We construct two molecular dynamics datasets using two proteins, i.e., *5AWL*, *2N5C*, and our approach is evaluated on the two datasets. Each sample in both datasets comprises a trajectory of molecular dynamics simulation, where the motions of each atom are governed by the Langevin dynamics equation in a specific solvent environment. The graph is constructed by comparing pairwise distance with a threshold, which would be updated at set intervals. The system parameters of the solvent are varied among different simulation samples. We target at predicting the position of every atom in three coordinates, i.e., $q_x$, $q_y$ and $q_z$. More details can be found in Appendix F.

**Performance Comparison.** We demonstrate the performance with respect to different prediction lengths in Table 2. From the results, we can conclude that our PGODE can achieve the best performance on two datasets in both ID and OOD settings. Note that molecular dynamics involves hundreds of atoms with complicated interacting rules. Therefore, the performance further demonstrates the strong expressivity of our PGODE for modeling challenging underlying rules.

**Visualization.** In addition, we provide the visualization of the two baselines and our PGODE in comparison to the ground truth with different prediction lengths in Figure 3. We can observe that our PGODE is capable of exploring more accurate dynamical patterns compared with the ground truth. More importantly, our PGODE can almost recover the position patterns when the prediction length is 24, which validates the capability of PGODE to handle complicated scenarios.

Table 3: Ablation study on *Springs* (MSE $\times 10^{-2}$) and *5AWL* (MSE $\times 10^{-3}$) with a prediction length of 24.

| Dataset | *Springs* (ID) | | *Springs* (OOD) | | *5AWL* (ID) | | | *5AWL* (OOD) | | |
|---|---|---|---|---|---|---|---|---|---|---|
| Variable | $q$ | $v$ | $q$ | $v$ | $q_x$ | $q_y$ | $q_z$ | $q_x$ | $q_y$ | $q_z$ |
| PGODE w/o O | 0.106 | 0.326 | 0.127 | 0.339 | 2.995 | 3.532 | 2.932 | 3.649 | 4.469 | 3.639 |
| PGODE w/o S | 0.089 | 0.397 | 0.124 | 0.417 | 2.935 | 3.612 | 3.034 | 3.538 | 4.541 | 3.741 |
| PGODE w/o F | 0.164 | 0.517 | 0.202 | 0.577 | 3.157 | 3.629 | 3.326 | 3.634 | 4.604 | 3.917 |
| PGODE w/o D | 0.073 | 0.296 | 0.091 | 0.348 | 3.077 | 3.453 | 2.961 | 3.684 | 4.399 | 3.623 |
| PGODE (Full Model) | **0.070** | **0.262** | **0.088** | **0.291** | **2.910** | **3.384** | **2.904** | **3.374** | **4.334** | **3.615** |

Figure 4: (a), (b) Performance with respect to different condition lengths on *Springs* and *5AWL*. (c) (d) Performance and running time with respect to different numbers of prototypes.

## 4.3 FURTHER ANALYSIS

**Ablation Study.** We introduce three model variants as follows: (1) *PGODE w/o O*, which removes the object-level contexts and only utilizes system-level contexts for $\boldsymbol{w}_i$; (2) *PGODE w/o S*, which removes the system-level contexts and only utilizes object-level contexts for $\boldsymbol{w}_i$; (3) *PGODE w/o F*, which merely adopts one prototype for graph ODE. (4) *PGODE w/o F*, which remove the disentanglement loss. From the results in Table 3, we can have several observations. *Firstly*, removing either object-level or system-level contexts would obtain worse performance, which validates that both contexts are crucial to determining the interacting patterns. *Secondly*, our full model achieves better performance compared with *PGODE w/o F*, which validates that different prototypes can increase the representation capacity for modeling complicated dynamics. *Thirdly*, in comparison to *PGODE w/o F* and the full model, we can infer that representation disentanglement greatly enhances the performance under system changes. More model variants can be found in Sec. H.2.

**Parameter Sensitivity.** We first analyze the influence of different conditional lengths and prediction lengths by varying them in $\{3, 6, 9, 12, 15\}$ and $\{12, 24\}$, respectively. As shown in Figure 4 (a) and (b), we can observe that the error would decrease till saturation as the condition length rises since more historical information is provided. In addition, PGODE can always perform better than HOPE in every setting. Then, we vary the number of prototypes in $\{2, 3, 4, 5, 6\}$ in Figure 4 (c) and observe that more prototypes would bring in better results before saturation.

**Efficiency.** Although more prototypes tend to benefit the performance, they can also bring in high computational cost. We show the computational time with respect to different numbers of prototypes in Figure 4 (d) and observe that the computational complexity would increase with more prototypes. Due to the trade-off between effectiveness and efficiency, we would set the number as 5.

## 5 CONCLUSION

In this work, we investigate a long-standing problem of modeling interacting dynamical systems and develop a novel approach named PGODE, which infers prototype decomposition from contextual discovery for a graph ODE framework. In particular, PGODE extracts disentangled object-level and system-level contexts from historical trajectories, which can enhance the capability of generalization under system changes. In addition, we present a graph ODE framework that determines a combination of multiple interacting prototypes for increased model expressivity. Extensive experiments demonstrate the superiority of the proposed PGODE in different settings in comparison with various competing approaches. In future work, we plan to extend our proposed PGODE with more advanced graph inference for more complicated scenarios.

ETHICS STATEMENT

We acknowledge that all co-authors of this work have read and committed to adhering to the ICLR Code of Ethics.

REPRODUCIBILITY STATEMENT

We have included all details about the datasets and our experiment settings in Appendix F and Appendix G, respectively. The anonymous source code can be found in `https://anonymous. 4open.science/r/GOAT/`. We will also make the code public to facilitate future research.

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

# A    PROOF OF LEMMA 3.1

**Lemma 4.1.** We first assume that the learnt functions $\psi_r^k : \mathbb{R}^{2d} \to \mathbb{R}^d, \psi_a^k : \mathbb{R}^d \to \mathbb{R}^d$ have bounded gradients. In other words, there exists $A, R > 0$, such that the following Jacobian matrices have bounded matrix norm:

$$J_{\psi_r^k}(\boldsymbol{x}, \boldsymbol{y}) = \begin{pmatrix} \frac{\partial \psi_{r,1}^k}{\partial x_1} & \cdots & \frac{\partial \psi_{r,1}^k}{\partial x_d} & \frac{\partial \psi_{r,1}^k}{\partial y_1} & \cdots & \frac{\partial \psi_{r,1}^k}{\partial y_d} \\ \vdots & \ddots & \vdots & \vdots & \ddots & \vdots \\ \frac{\partial \psi_{r,d}^k}{\partial x_1} & \cdots & \frac{\partial \psi_{r,d}^k}{\partial x_d} & \frac{\partial \psi_{r,d}^k}{\partial y_1} & \cdots & \frac{\partial \psi_{r,d}^k}{\partial y_d} \end{pmatrix}, \quad \|J_{\psi_r^k}(\boldsymbol{x}, \boldsymbol{y})\| \le R, \tag{17}$$

$$J_{\psi_a^k}(\boldsymbol{x}) = \begin{pmatrix} \frac{\partial \psi_{a,1}^k}{\partial x_1} & \cdots & \frac{\partial \psi_{a,1}^k}{\partial x_d}, \\ \vdots & \ddots & \vdots \\ \frac{\partial \psi_{a,d}^k}{\partial x_1} & \cdots & \frac{\partial \psi_{a,d}^k}{\partial x_d} \end{pmatrix}, \quad \|J_{\psi_a^k}(\boldsymbol{x})\| \le A. \tag{18}$$

Then, given the initial state $(t_0, \boldsymbol{z}_1^{t_0}, \cdots, \boldsymbol{z}_N^{t_0}, \boldsymbol{w}_1, \cdots, \boldsymbol{w}_N)$, we claim that there exists $\varepsilon > 0$, such that the ODE system Eqn. 10 has a unique solution in the interval $[t_0 - \varepsilon, t_0 + \varepsilon]$.

We first introduce the Picard–Lindelöf Theorem as below.

**Theorem A.1.** *(Picard–Lindelöf Theorem) Let $D \subseteq \mathbb{R} \times \mathbb{R}^n$ be a closed rectangle with $(t_0, y_0) \in D$. Let $f : D \to \mathbb{R}^n$ be a function that is continuous in $t$ and Lipschitz continuous in $y$. Then, there exists some $\varepsilon > 0$ such that the initial value problem:*

$$y'(t) = f(t, y(t)), \quad y(t_0) = y_0. \tag{19}$$

*has a unique solution $y(t)$ on the interval $[t_0 - \varepsilon, t_0 + \varepsilon]$.*

Then, we prove the following lemma.

**Lemma A.1.** *Suppose we have a series of L-Lipschitz continuous functions $\{f_i : \mathbb{R}^m \to \mathbb{R}^n\}_{i=1}^N$, and then their linear combination is also L-Lipschitz continuous, i.e., $\forall \{a_1, \cdots a_N\} \in [0, 1]^N$, satisfying $\sum_{i=1}^N a_i = 1$, we have $\sum_{i=1}^N a_i f_i$ is also L-Lipschitz continuous.*

*Proof.* $\forall \boldsymbol{x}, \boldsymbol{y} \in \mathbb{R}^m$, we have:

$$\|\sum_{i=1}^N a_i f_i(\boldsymbol{x}) - \sum_{i=1}^N a_i f_i(\boldsymbol{y})\| \le \sum_{i=1}^N a_i \|f_i(\boldsymbol{x}) - f_i(\boldsymbol{y})\| \tag{20}$$

$$\le \sum_{i=1}^N a_i L \|\boldsymbol{x} - \boldsymbol{y}\| \tag{21}$$

$$= L \|\boldsymbol{x} - \boldsymbol{y}\|. \tag{22}$$

$\square$

Next, we show the proof of Lemma 3.1.

*Proof.* First, we can rewrite the ODE system Eqn. 10 as:

$$\frac{d\boldsymbol{Z}^t}{dt} = \sum_{k=1}^K \boldsymbol{W}^k f^k(\boldsymbol{Z}^t) - \boldsymbol{Z}^t, \tag{23}$$

where $\boldsymbol{W}^k \in \mathbb{R}^{Nd \times Nd}$ is a diagonal matrix. It is evident that the right hand side is continuous with respect to $t$ since it does not depend on $t$ directly.

Then, for any continuous function $f : \mathbb{R}^n \to \mathbb{R}^m$, with the Mean Value Theorem, we have $\forall \boldsymbol{x}, \boldsymbol{y} \in \mathbb{R}^n, \|f(\boldsymbol{x}) - f(\boldsymbol{y})\| = \|J_f(\boldsymbol{p})\| * \|\boldsymbol{x} - \boldsymbol{y}\|$, where $\boldsymbol{p}$ is a point in the segment connecting $\boldsymbol{x}$ and $\boldsymbol{y}$.

Now, denote $A(i,j) \in \mathbb{R}^{2 \times dN}$ with the first row has elements with index from idN+1 to (i+1)dN be 1, the others 0; the second row has elements with index from jdN+1 to (j+1)dN be 1, the others 0.

By introducing $A(i,j)$, for all $\boldsymbol{X} = \begin{pmatrix} \boldsymbol{x}_1 \\ \vdots \\ \boldsymbol{x}_N \end{pmatrix}, \boldsymbol{Y} = \begin{pmatrix} \boldsymbol{y}_1 \\ \vdots \\ \boldsymbol{y}_N \end{pmatrix} \in \mathbb{R}^{dN}$, we have:

$$\|\psi_r^k(A(i,j)\boldsymbol{X}) - \psi_r^k(A(i,j)\boldsymbol{Y})\| \leq \|\psi_r^k(\boldsymbol{x}_i, \boldsymbol{x}_j) - \psi_r^k(\boldsymbol{y}_i, \boldsymbol{x}_j)\| + \|\psi_r^k(\boldsymbol{y}_i, \boldsymbol{x}_j) - \psi_r^k(\boldsymbol{y}_i, \boldsymbol{y}_j)\| \tag{24}$$

$$= \|J_{\psi_r^k}(\boldsymbol{p}_i)\| * \|\boldsymbol{x}_i - \boldsymbol{y}_i\| + \|J_{\psi_r^k}(\boldsymbol{p}_j)\| * \|\boldsymbol{x}_j - \boldsymbol{y}_j\| \tag{25}$$

$$\leq R\|\boldsymbol{x}_i - \boldsymbol{y}_i\| + R\|\boldsymbol{x}_j - \boldsymbol{y}_j\| \tag{26}$$

$$\leq R\|\boldsymbol{X} - \boldsymbol{Y}\|, \tag{27}$$

where $\boldsymbol{p}_i$ is a point in the segment connecting $\boldsymbol{x}_i$ and $\boldsymbol{y}_i$, and a similar definition is for $\boldsymbol{p}_j$. Note that we have $\psi_r^k$ is R-Lipschitz continuous. Therefore, by Lemma A.1, the following linear combination is also R-Lipschitz continuous:

$$l^k(\boldsymbol{Z}^t) = \sum_{j^t \in \mathcal{S}(i^t)} \psi_r^k([A(i^t, j^t)\boldsymbol{Z}^t]). \tag{28}$$

Thus, for all $\boldsymbol{X}, \boldsymbol{Y} \in \mathbb{R}^{dN}$, we have:

$$\|f^k(\boldsymbol{X}) - f^k(\boldsymbol{Y})\| = \|\psi_a^k(l^k(\boldsymbol{X})) - \psi_a^k(l^k(\boldsymbol{Y}))\| \tag{29}$$

$$\leq A\|l^k(\boldsymbol{X}) - l^k(\boldsymbol{Y})\| \tag{30}$$

$$\leq AR\|\boldsymbol{X} - \boldsymbol{Y}\|. \tag{31}$$

Again, we have each $f^k$ is AR-Lipschitz continuous, so their linear combination $\sum_{k=1}^K \boldsymbol{W}^k f^k$ will also be AR-Lipschitz continuous. Finally, we have

$$\|[\sum_{k=1}^K \boldsymbol{W}^k f^k(\boldsymbol{X}) - \boldsymbol{X}] - [\sum_{k=1}^K \boldsymbol{W}^k f^k(\boldsymbol{Y}) - \boldsymbol{Y}]\| \leq \|\sum_{k=1}^K \boldsymbol{W}^k f^k(\boldsymbol{X}) - \sum_{k=1}^K \boldsymbol{W}^k f^k(\boldsymbol{Y})\| \tag{32}$$

$$+ \|\boldsymbol{X} - \boldsymbol{Y}\| \tag{33}$$

$$\leq (AR+1)\|\boldsymbol{X} - \boldsymbol{Y}\|. \tag{34}$$

Thus, the right hand side will be (AR+1)-Lipschitz continuous. According to the Theorem A.1, we prove the uniqueness of the solution to Eqn. 10. $\square$

## B RELATED WORK

### B.1 INTERACTING DYNAMICS MODELING

Recent years have witnessed a surge of interest in modeling interacting dynamical systems across a variety of fields including molecular biology and computational physics (Shao et al., 2022; Lan et al., 2022; Li et al., 2022b; Bishnoi et al., 2022). While convolutional neural networks (CNNs) have been successfully employed to learn from regular data such as grids and frames (Peng et al., 2020), emerging research is increasingly utilizing geometric graphs to represent more complex systems (Wu et al., 2023; Deng et al., 2023). Graph neural networks (GNNs) have thus become increasingly prevailing for modeling these intricate dynamics (Pfaff et al., 2021; Shao et al., 2022; Sanchez-Gonzalez et al., 2020; Allen et al., 2022; Look et al., 2023; Yıldız et al., 2022). AgentFormer (Yuan et al., 2021) jointly models both time and social dimensions with semantic information preserved. NRI (Kipf et al., 2018) models interactions along with node states from observations using GNNs. R-SSM (Yang et al., 2020) models the dynamics of interacting objects using GNNs and includes auxiliary contrastive prediction tasks to enhance discriminative learning. Despite their popularity, current methods often fall short in modeling challenging scenarios such as out-of-distribution shift and long-term dynamics (Yu et al., 2021). To address these limitations, our work leverages contextual knowledge to incorporate prototype decomposition into a graph ODE framework.

### B.2 Neural Ordinary Differential Equations

Motivated by the approximation of residual networks (Chen et al., 2018), neural ordinary differential equations (ODEs) have been introduced to model continuous-time dynamics using parameterized derivatives in hidden spaces. These neural ODEs have found widespread use in time-series forecasting due to their effectiveness (Dupont et al., 2019; Xia et al., 2021; Jin et al., 2022; Schirmer et al., 2022). Incorporated with the message passing mechanism, they have been integrated with GNNs, which can mitigate the issue of oversmoothing and enhance model interpretability (Xhonneux et al., 2020; Zhang et al., 2022; Poli et al., 2019). I-GPODE (Yıldız et al., 2022) estimates the uncertainty of trajectory predictions using the Gaussian process, which facilitates effective long-term predictions. HOPE (Luo et al., 2023) focuses on incorporating second-order graph ODE in evolution modeling. In contrast, our method introduces hierarchical context discovery with disentanglement to guide the prototype decomposition of individual nodes in modeling interacting dynamics.

### B.3 Graph Neural Networks

Graph Neural Networks (GNNs) (Kipf & Welling, 2017; Xu et al., 2019a; Veličković et al., 2018) have shown remarkable efficacy in handling a range of graph-based machine learning tasks such as node classification (Yang et al., 2021) and graph classification (Liu et al., 2022). Typically, they adopt the message passing mechanism, where each node aggregates messages from its adjacent nodes for updated node representations. Recently, researchers have started to focus more on realistic graphs that do not obey the homophily assumption and developed several GNN approaches to tackle heterophily (Zhu et al., 2021; Li et al., 2022a; Zhu et al., 2020). These approaches typically leverage new graph structures (Zhu et al., 2020; Suresh et al., 2021) and modify the message passing procedures (Chien et al., 2021; Yan et al., 2022) to mitigate the influence of potential heterophily. In our PGODE, we focus on interacting dynamics systems instead. In particular, due to the local heterophily, different objects should have different interacting patterns, and therefore we infer object-level contexts from historical data.

## C More Discussion About Expressivity

We provide more discussion about the expressivity of the proposed PGODE. Piecewise continuous neural networks have been proven asymptotically more expressive than classical feed forward networks (Kratsios & Zamanlooy, 2022). Our prototype decomposition adopts a soft form of piecewise functions to enhance the expressivity, which can also help capture the influence of seasonality and events in real-world dynamics systems. Our empirical results in ID settings also validate the strong expressivity when handling complicated dynamics.

## D Algorithm

We summarize the learning algorithm of our PGODE in Algorithm 1.

## E Detail of Baselines

Our approach is compared with various baselines for dynamics systems modeling, i.e., LSTM (Hochreiter & Schmidhuber, 1997), GRU (Weerakody et al., 2021), NODE (Chen et al., 2018), LG-ODE (Huang et al., 2020), MPNODE (Chen et al., 2022), SocialODE (Wen et al., 2022) and HOPE (Luo et al., 2023).

The proposed method is compared with seven competing baselines as follows:

- LSTM (Hochreiter & Schmidhuber, 1997) has been broadly utilized for sequence prediction tasks. Compared with classic RNNs, LSTM incorporates three critical gates, i.e., the forget gate, the input gate, and the output gate, which can effectively understand and retain important long-term dependencies within the data sequences.

---

**Algorithm 1** Training Algorithm of PGODE

---

**Input:** The observations $G^{1:T} = \{G^1, \cdots, G^T\}$.
**Output**: The parameters in our model.

---

1: Initialize model parameters;
2: **while** not convergence **do**
3:     **for** each training sequence **do**
4:         Partition the sequence into two segments;
5:         Construct the temporal graph using Eqn. 2;
6:         Generate object-level contexts using Eqn. 5;
7:         Generate system-level contexts with summarization;
8:         Solve our prototypical graph ODE in Eqn. 10;
9:         Output the trajectories using the decoder;
10:         Calculate the final loss in Eqn. 16;
11:         Update $\tau'$ in our PGODE using gradient ascent;
12:         Update other parameters in our PGODE using gradient descent;
13:     **end for**
14: **end while**

---

- GRU (Cho et al., 2014) is another popular RNN architecture, which employs the gating mechanism to control the information flow during propagation. GRU has an improved computational efficiency compared LSTM.

- NODE (Chen et al., 2018) is the first method to introduce a continuous neural network based on the residual connection. It has been shown effective in time-series forecasting.

- LG-ODE (Huang et al., 2020) incorporates graph neural networks with neural ODE, which can capture continuous interacting dynamics in irregularly-sampled partial observations.

- MP-NODE (Gupta et al., 2022) combines the message passing mechanism and neural ODEs, which can capture sub-system relationships during the evolution of homogeneous systems.

- SocialODE (Wen et al., 2022) simulates the evolution of agent states and their interactions using a neural ODE architecture, which shows remarkable performance in multi-agent trajectory forecasting.

- HOPE (Luo et al., 2023) is a recently proposed graph ODE method, which adopts a twin encoder to learn latent state representations. These representations are fed into a high-order graph ODE to learn long-term correlations from complicated dynamical systems.

## F   DATASET DETAILS

We use four simulation datasets to evaluate our proposed GOAT, including physical and molecular dynamic systems. We will introduce the details of these four datasets in this part.

- *Springs* & *Charged*. The two physical dynamic simulation datasets *Springs* and *Charged* are commonly used in the field of machine learning for simulating physical systems. The *Springs* dataset simulates a system of interconnected springs governed by Hooke's law. Each spring has inherent properties such as elasticity coefficients and initial positions, representing a dynamic mechanical system. Each sample in the *Springs* dataset contains 10 interacting springs with information about the current state, i.e., velocity and acceleration, and additional properties, i.e., mass and damping coefficients. Similar to the *Springs* dataset, *Charged* is another popular physical dynamic simulation dataset that simulates electromagnetic phenomena. The objects in *Charged* are replaced by the electronics. We use the box size $\alpha$, the initial velocity $\beta$, the interaction strength $\gamma$, and springcharged probability $\delta$ as the system parameters in the experiments. It is noteworthy that the objects attract or repel with equal probability in the *Charged* system but unequal probability in the spring system. Both systems have a given graph indicating fixed interactions from real springs or electric charge effects.

- *5AWL & 2N5C.* To evaluate our approach on modeling molecular dynamic systems, we construct two datasets from two proteins, *5AWL* and *2N5C*, which can be accessed from the RCSB[1]. First, we repair missing residues, non-standard residues, missing atoms, and hydrogen atoms in the selected protein. Additionally, we adjust the size of the periodic boundary box to ensure that it is sufficiently large, thus avoiding truncation effects and abnormal behavior of the simulation system during the data simulation process. Then, we perform simulations on the irregular molecular motions within the protein using Langevin Dynamics (García-Palacios & Lázaro, 1998) under the NPT (isothermal-isobaric ensemble) conditions, with parameters sampled from the specified range, and we extract a frame every $0.2\ ps$ to record the protein structure, which constitutes the dataset used for supervised learning. In the two constructed datasets, we use the temperature $t$, pressure value $p$, and frictional coefficient $\mu$ as the dynamic system parameters. Langevin Dynamics is a mathematical model used to simulate the flow dynamics of molecular systems (Bussi & Parrinello, 2007). It can simplify complex systems by replacing some degrees of freedom of the molecules with stochastic differential equations. For a dynamic system containing $N$ particles of mass $m$, with coordinates given by $X = X(t)$, the Langevin equation of it can be formulated as follows:

$$m\frac{d^2 X}{dt^2} = -\Delta U(X) - \mu\frac{dX}{dt} + \sqrt{2\mu k_b T}R(t), \tag{35}$$

where $\mu$ represents the frictional coefficient, $\Delta U(X)$ is the interaction potential between particles, $\Delta$ is the gradient operator, $T$ is the temperature, $k_b$ is Boltzmann constant and $R(t)$ is delta-correlated stationary Gaussian process.

## G IMPLEMENTATION DETAILS

Table 4: Datasets and distributions of system parameters. For the OOD test set, there is at least one of the system parameters outside the range utilized for training. $\alpha$: box size, $\beta$: initial velocity norm, $\gamma$: interaction strength, $\delta$: spring/charged probability. $t$: temperature, $p$: pressure, $\mu$: frictional coefficient.

|  | *Springs* | *Charged* | *5AWL/2N5C* |
|---|---|---|---|
| Parameters | $\alpha, \beta, \gamma, \delta$ | $\alpha, \beta, \gamma, \delta$ | $t, p, \mu$ |
| Train/Val/Test | $A = \{\alpha \in [4.9, 5.1]\}$
$B = \{\beta \in [0.49, 0.51]\}$
$C = \{\gamma \in [0.09, 0.11]\}$
$D = \{\delta \in [0.49, 0.51]\}$
$\Omega_{\text{train}} = (A \times B \times C \times D)$ | $A = \{\alpha \in [4.9, 5.1]\}$
$B = \{\beta \in [0.49, 0.51]\}$
$C = \{\gamma \in [0.9, 1.1]\}$
$D = \{\delta \in [0.49, 0.51]\}$
$\Omega_{\text{train}} = (A \times B \times C \times D)$ | $T = \{t \in [290, 310]\}$
$P = \{p \in [0.9, 1.1]\}$
$M = \{\mu \in [0.9, 1.1]\}$
$\Omega_{\text{train}} = (T \times P \times M)$ |
| OOD Test Set | $A = \{\alpha \in [4.8, 5.2]\}$
$B = \{\beta \in [0.48, 0.52]\}$
$C = \{\gamma \in [0.08, 0.12]\}$
$D = \{\delta \in [0.48, 0.52]\}$
$\Omega_{\text{OOD}} =$
$(A \times B \times C \times D) \setminus \Omega_{\text{train}}$ | $A = \{\alpha \in [4.8, 5.2]\}$
$B = \{\beta \in [0.48, 0.52]\}$
$C = \{\gamma \in [0.8, 1.2]\}$
$D = \{\delta \in [0.48, 0.52]\}$
$\Omega_{\text{OOD}} =$
$(A \times B \times C \times D) \setminus \Omega_{\text{train}}$ | $T = \{t \in [280, 320]\}$
$P = \{p \in [0.8, 1.2]\}$
$M = \{\mu \in [0.8, 1.2]\}$
$\Omega_{\text{OOD}} =$
$(T \times P \times M) \setminus \Omega_{\text{train}}$ |
| Number of samples | | | |
| Train/Val/Test | 1000/200/200 | | 200/50/50 |
| OOD Test Set | 200 | | 50 |

In our experiments, we employ a rigorous data split strategy to ensure the accuracy of our results. Specifically, we split the whole datasets into four different parts, including the normal three sets, i.e., training, validating and in-distribution (ID) test sets and an out-of-distribution (OOD) test set. For the physical dynamic datasets, we generate 1200 samples for training and validating, 200 samples for ID testing and 200 samples for OOD testing. For the molecular dynamic datasets, we construct 200 samples for training, 50 samples for validating, 50 samples for ID testing and 50 samples for testing in OOD settings.

Each sample in the datasets has a group of distinct system parameters as shown in Table 4. For training, validation and ID test samples, we randomly sample system parameters in the space of

---

[1]https://www.rcsb.org

$\Omega_{train}$. For OOD samples, the system parameters come from $\Omega_{OOD}$ randomly, which indicates distribution shift compared with the training domain. During the training process, each trajectory sample is further split into two parts, i.e., a conditional part for initializing object-level contexts representation and global-level contexts representation, and a prediction part for supervising the model. The size of the two parts is denoted as conditional length and prediction length, respectively. In our experiments, we set the conditional length to 12, and we used three different prediction lengths, i.e., 12, 24, and 36.

We adopt PyTorch (Paszke et al., 2017) and torchdiffeq package (Kidger et al., 2021) to implement all the compared approaches and our PGODE. All these experiments in this work are performed on a single NVIDIA A40 GPU. The fourth-order Runge-Kutta method from torchdiffeq is adopted as the ODE solver. We employ a set of one-layer GNN prototypes with a hidden dimension of 128 for graph ODE. The number of prototypes is set to 5 as default. For optimization, we utilize an Adam optimizer (Kingma & Ba, 2015) with an initial learning rate of 0.0005. The batch size is set to 256 for the physical dynamic simulation datasets and 64 for the molecular dynamic simulation datasets.

## H    MORE EXPERIMENT RESULTS

### H.1    PERFORMANCE COMPARISON

To begin, we compare with our PGODE with more baselines, i.e., AgentFormer (Yuan et al., 2021), NRI (Kipf et al., 2018) and I-GPODE (Yıldız et al., 2022) in our performance comparison. The results of these comparisons are presented in Table 5 and our method outperforms the compared methods. In addition, we show the performance of the compared methods in two different coordinates of positions and velocities, i.e., $q_x$, $q_y$, $v_x$ and $v_y$. The compared results on *Springs* and *Charged* are shown in Table 6 and Table 7, respectively. From the results, we can observe the superiority of the proposed PGODE in capturing complicated interacting patterns under both ID and OOD settings.

Table 5: Performance comparison with NRI, AgentFormer, and I-GPODE on physical dynamics simulations (MSE $\times 10^{-2}$). NRI, AgentFormer, and I-GPODE are out of memory on molecular dynamics simulations.

| Prediction Length | 12 (ID) | | 24 (ID) | | 36 (ID) | | 12 (OOD) | | 24 (OOD) | | 36 (OOD) | |
|---|---|---|---|---|---|---|---|---|---|---|---|---|
| Variable | $q$ | $v$ | $q$ | $v$ | $q$ | $v$ | $q$ | $v$ | $q$ | $v$ | $q$ | $v$ |
| *Springs* | | | | | | | | | | | | |
| NRI | 0.103 | 0.425 | 0.210 | 0.681 | 0.693 | 2.263 | 0.119 | 0.472 | 0.246 | 0.770 | 0.807 | 2.406 |
| AgentFormer | 0.115 | 0.163 | 0.202 | 0.517 | 1.656 | 1.691 | 0.157 | 0.195 | 0.243 | 0.505 | 1.875 | 1.913 |
| I-GPODE | 0.159 | 0.479 | 0.746 | 3.002 | 1.701 | 7.433 | 0.173 | 0.498 | 0.796 | 3.193 | 1.818 | 7.322 |
| PGODE (Ours) | **0.035** | **0.124** | **0.070** | **0.262** | **0.296** | **1.326** | **0.047** | **0.138** | **0.088** | **0.291** | **0.309** | **1.337** |
| *Charged* | | | | | | | | | | | | |
| NRI | 0.901 | 2.702 | 3.225 | 3.346 | 7.770 | 4.543 | 1.303 | 2.726 | 3.678 | 3.548 | 8.055 | 4.752 |
| AgentFormer | 1.076 | 2.476 | 3.631 | 3.044 | 7.513 | 3.944 | 1.384 | 2.514 | 4.224 | 3.199 | 8.985 | 4.002 |
| I-GPODE | 1.044 | 2.818 | 3.407 | 3.751 | 7.292 | 4.570 | 1.322 | 2.715 | 3.805 | 3.521 | 8.011 | 4.056 |
| PGODE (Ours) | **0.578** | **2.196** | **2.037** | **2.648** | **4.804** | **3.551** | **0.802** | **2.135** | **2.584** | **2.663** | **5.703** | **3.703** |

### H.2    ABLATION STUDY

We show more ablation studies on *Charged* and *2N5C* to make our analysis complete. In particular, the compared performance of different model variants are shown in Table 8. From the results, we can observe that our full model can outperform all the model variance in all cases, which validates the effectiveness of each component in our PGODE again. In addition, we introduce two model variants: (1) PGODE w. MLP, which combines a GNN with an MLP to learn the individualized dynamics; (2) PGODE w. Single, which takes the node representation and the global representation as input with a single message passing function. The compared performance of different model variants is shown in Table 9. From the results, we can observe that our full model can outperform all the model variance in all cases. Compared with these variants, our prototype decomposition can involve different GNN bases, which model diverse evolving patterns to jointly determine the individualized dynamics. This strategy can enhance the model expressivity, allowing for more accurate representation learning of hierarchical structures from a mixture-of-experts perspective.

Table 6: Mean Squared Error (MSE) $\times 10^{-2}$ on *Springs*.

| Prediction Length | 12 | | | | 24 | | | | 36 | | | |
|---|---|---|---|---|---|---|---|---|---|---|---|---|
| Variable | $q_x$ | $q_y$ | $v_x$ | $v_y$ | $q_x$ | $q_y$ | $v_x$ | $v_y$ | $q_x$ | $q_y$ | $v_x$ | $v_y$ |
| *ID* | | | | | | | | | | | | |
| LSTM | 0.324 | 0.250 | 0.909 | 0.931 | 0.679 | 0.638 | 2.695 | 2.623 | 1.253 | 1.304 | 5.023 | 6.434 |
| GRU | 0.496 | 0.291 | 0.565 | 0.628 | 0.873 | 0.623 | 1.711 | 2.001 | 1.368 | 1.128 | 2.980 | 3.912 |
| NODE | 0.165 | 0.148 | 0.649 | 0.479 | 0.722 | 0.621 | 2.534 | 2.293 | 1.683 | 1.534 | 6.323 | 6.142 |
| LG-ODE | 0.077 | 0.077 | 0.264 | 0.272 | 0.174 | 0.135 | 0.449 | 0.576 | 0.613 | 0.441 | 1.757 | 2.528 |
| MPNODE | 0.080 | 0.072 | 0.222 | 0.263 | 0.237 | 0.105 | 0.407 | 0.506 | 0.866 | 0.335 | 1.469 | 2.006 |
| SocialODE | 0.069 | 0.068 | 0.205 | 0.315 | 0.138 | 0.120 | 0.391 | 0.630 | 0.429 | 0.400 | 1.751 | 2.624 |
| HOPE | 0.087 | 0.053 | 0.152 | 0.200 | 0.571 | 0.342 | 0.707 | 1.206 | 2.775 | 2.175 | 4.412 | 6.405 |
| PGODE (Ours) | **0.033** | **0.037** | **0.122** | **0.127** | **0.074** | **0.066** | **0.239** | **0.286** | **0.318** | **0.273** | **1.186** | **1.466** |
| *OOD* | | | | | | | | | | | | |
| LSTM | 0.499 | 0.449 | 1.086 | 1.227 | 1.019 | 0.857 | 2.847 | 2.466 | 1.768 | 1.415 | 5.154 | 5.293 |
| GRU | 0.714 | 0.469 | 0.713 | 0.703 | 1.280 | 0.905 | 1.795 | 2.096 | 1.844 | 1.497 | 2.852 | 3.994 |
| NODE | 0.246 | 0.209 | 0.997 | 0.585 | 0.876 | 0.687 | 2.790 | 2.269 | 2.002 | 1.663 | 6.349 | 5.670 |
| LG-ODE | 0.093 | 0.083 | 0.272 | 0.327 | 0.185 | 0.172 | 0.463 | 0.661 | 0.684 | 0.545 | 1.767 | 2.645 |
| MPNODE | 0.107 | 0.081 | 0.230 | 0.268 | 0.299 | 0.126 | 0.420 | 0.528 | 0.967 | 0.386 | 1.464 | 1.969 |
| SocialODE | 0.082 | 0.076 | 0.221 | 0.350 | 0.151 | 0.156 | 0.414 | 0.726 | 0.488 | 0.495 | 1.793 | 2.826 |
| HOPE | 0.094 | 0.058 | 0.178 | 0.264 | 0.506 | 0.523 | 1.031 | 1.603 | 2.369 | 2.251 | 3.701 | 8.291 |
| PGODE (Ours) | **0.046** | **0.048** | **0.133** | **0.144** | **0.094** | **0.081** | **0.286** | **0.297** | **0.336** | **0.281** | **1.360** | **1.313** |

Table 7: Mean Squared Error (MSE) $\times 10^{-2}$ on *Charged*.

| Prediction Length | 12 | | | | 24 | | | | 36 | | | |
|---|---|---|---|---|---|---|---|---|---|---|---|---|
| Variable | $q_x$ | $q_y$ | $v_x$ | $v_y$ | $q_x$ | $q_y$ | $v_x$ | $v_y$ | $q_x$ | $q_y$ | $v_x$ | $v_y$ |
| *ID* | | | | | | | | | | | | |
| LSTM | 0.743 | 0.846 | 2.913 | 3.145 | 2.797 | 3.052 | 3.605 | 3.863 | 6.477 | 6.660 | 4.240 | 4.423 |
| GRU | 0.764 | 0.799 | 2.931 | 3.063 | 2.709 | 2.901 | 3.572 | 3.709 | 5.657 | 6.281 | 4.068 | 4.227 |
| NODE | 0.743 | 0.808 | 2.764 | 2.777 | 2.913 | 3.114 | 3.432 | 3.451 | 6.468 | 6.868 | 3.997 | 4.089 |
| LG-ODE | 0.736 | 0.783 | 2.322 | 2.414 | 2.320 | 2.731 | 3.361 | 3.268 | 5.188 | 6.782 | 6.194 | 5.043 |
| MPNODE | 0.720 | 0.759 | 2.414 | 2.496 | 2.379 | 2.536 | 3.589 | 3.738 | 5.636 | 5.614 | 5.472 | 7.046 |
| SocialODE | 0.630 | 0.695 | 2.311 | 2.358 | 2.252 | 2.631 | 3.509 | 2.995 | 5.743 | 7.076 | 5.701 | 4.122 |
| HOPE | 0.593 | 0.635 | 2.295 | 2.337 | 3.214 | 2.938 | 3.279 | 3.482 | 9.289 | 7.845 | 8.406 | 8.511 |
| PGODE (Ours) | **0.555** | **0.600** | **2.164** | **2.228** | **1.940** | **2.134** | **2.624** | **2.673** | **4.449** | **5.159** | **3.778** | **3.324** |
| *OOD* | | | | | | | | | | | | |
| LSTM | 1.130 | 1.123 | 3.062 | 2.992 | 4.026 | 3.950 | 3.768 | 3.512 | 7.934 | 8.435 | 4.517 | 3.925 |
| GRU | 1.072 | 1.012 | 3.108 | 2.948 | 3.893 | 3.602 | 3.844 | 3.428 | 6.970 | 8.061 | 4.485 | 3.718 |
| NODE | 1.185 | 1.062 | 2.956 | 2.732 | 4.057 | 3.804 | 3.645 | 3.480 | 8.622 | 8.372 | 5.097 | 4.376 |
| LG-ODE | 0.999 | 0.866 | 2.581 | 2.521 | 2.797 | 3.239 | 4.200 | 2.978 | 5.996 | 7.593 | 8.422 | 4.309 |
| MPNODE | 1.092 | 0.897 | 2.487 | 2.623 | 2.967 | 2.828 | 3.670 | 4.001 | 6.051 | 6.118 | 6.029 | 7.566 |
| SocialODE | 0.865 | 0.924 | 2.481 | 2.359 | 2.610 | 3.177 | 3.968 | 2.836 | **5.482** | 7.102 | 8.530 | 4.150 |
| HOPE | 0.839 | 0.918 | 2.466 | 2.484 | 3.586 | 3.783 | 3.417 | 3.442 | 11.254 | 10.652 | 10.133 | 8.107 |
| PGODE (Ours) | **0.739** | **0.865** | **2.159** | **2.110** | **2.524** | **2.643** | **2.704** | **2.623** | 5.748 | **5.659** | **4.017** | **3.389** |

Table 8: Ablation study on *Charged* (MSE $\times 10^{-2}$) and *2N5C* (MSE $\times 10^{-3}$) with a prediction length of 24.

| Dataset | *Charged* (ID) | | *Charged* (OOD) | | *2N5C* (ID) | | | *2N5C* (OOD) | | |
|---|---|---|---|---|---|---|---|---|---|---|
| Variable | $q$ | $v$ | $q$ | $v$ | $q_x$ | $q_y$ | $q_z$ | $q_x$ | $q_y$ | $q_z$ |
| PGODE w/o O | 2.282 | 3.013 | 2.590 | 2.943 | 2.076 | 2.130 | 2.215 | 2.582 | 2.800 | 2.833 |
| PGODE w/o S | 2.308 | 2.994 | 2.990 | 2.911 | 2.040 | 2.046 | 2.227 | 2.559 | 2.791 | 2.854 |
| PGODE w/o F | 2.497 | 3.298 | 2.882 | 3.197 | 2.424 | 2.208 | 2.465 | 2.970 | 2.868 | 3.118 |
| PGODE w/o D | 2.179 | 2.842 | 2.616 | 3.076 | 2.119 | 2.083 | 2.171 | 2.785 | 2.759 | 2.829 |
| PGODE (Full Model) | **2.037** | **2.648** | **2.584** | **2.663** | **1.960** | **2.029** | **2.119** | **2.464** | **2.734** | **2.727** |

Table 9: Further ablation study on *Springs* (MSE $\times 10^{-2}$) and *5AWL* (MSE $\times 10^{-3}$) with a prediction length of 24.

| Dataset | *Springs* (ID) | | *Springs* (OOD) | | *5AWL* (ID) | | | *5AWL* (OOD) | | |
|---|---|---|---|---|---|---|---|---|---|---|
| Variable | $q$ | $v$ | $q$ | $v$ | $q_x$ | $q_y$ | $q_z$ | $q_x$ | $q_y$ | $q_z$ |
| PGODE w. Single | 0.208 | 0.434 | 0.248 | 0.481 | 3.010 | 3.741 | 3.143 | 3.523 | 4.691 | 3.839 |
| PGODE w. MLP | 0.152 | 0.454 | 0.179 | 0.514 | 2.997 | 3.638 | 3.240 | 3.605 | 4.492 | 3.908 |
| PGODE (Full Model) | **0.070** | **0.262** | **0.088** | **0.291** | **2.910** | **3.384** | **2.904** | **3.374** | **4.334** | **3.615** |

## H.3 PERFORMANCE WITH DIFFERENT NUMBER OF PROTOTYPES

Figure 5 (a) (b) (c) and (d) record the performance with respect to different numbers of prototypes on different datasets. From the results, we can find that more prototypes would bring in better results before saturation.

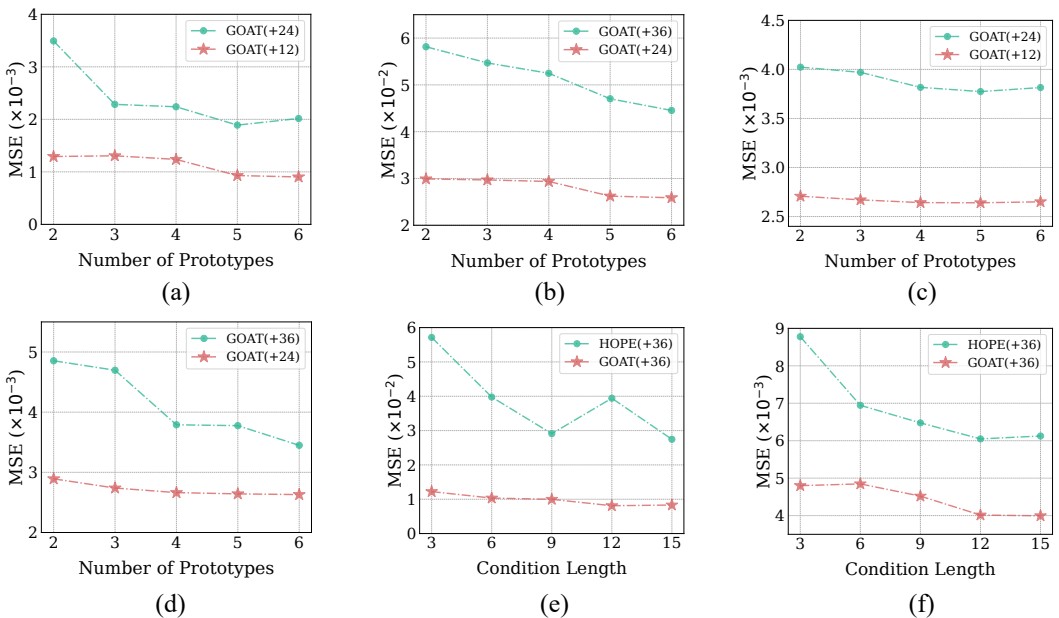

Figure 5: (a),(b),(c),(d) Performance on the OOD test set of *Springs*, *Charged*, *5AWL*, and *2N5C* with respect to four different numbers of prototypes. (e),(f) Performance with respect to different condition lengths on the ID test set of *Springs* and *5AWL*.

## H.4 PERFORMANCE WITH DIFFERENT CONDITION LENGTHS

We analyze the influence of different conditional lengths by varying them in $\{3, 6, 9, 12, 15\}$, respectively. As shown in Figure 5 (e) and (f), we can observe that our PGODE can always outperform the latest baseline HOPE, which validates the superiority of the proposed PGODE.

## H.5 EFFICIENCY COMPARISON

We have conducted a comparison of computation cost. The results are shown in Table 10 and we can observe that our method has a competitive computation cost. In particular, the performance of HOPE is much worse than ours (the increasement of ours is over 47% compared with HOPE), while our computational burden only increases a little. Moreover, both the performance and efficiency of I-GPODE are worse than ours.

Table 10: Comparison of training cost per epoch (s).

| Method | LSTM | GRU | NODE | LG-ODE | MPNODE | SocialODE | I-GPODE | HOPE | PGODE (Ours) |
|---|---|---|---|---|---|---|---|---|---|
| Springs | 1.53 | 1.04 | 2.21 | 17.39 | 23.33 | 21.02 | 267.08 | 23.86 | 37.03 |
| Charged | 1.33 | 1.02 | 2.06 | 16.59 | 22.26 | 19.93 | 250.23 | 20.43 | 33.88 |

## H.6 VISUALIZATION

In addition, we present more visualization of the proposed PGODE and two baselines, i.e., SocialODE and HOPE. We have offered visualization of the predicted trajectory of a sample in Figure 2 and now we visualize four extra test instances (two ID samples and two OOD samples) in Figure 6. From the results, we can observe that the proposed PGODE is capable of generating more reliable trajectories in comparison to the baselines. For instance, our PGODE can discover the correct direction of the orange particle while the others fail in the second OOD instance.

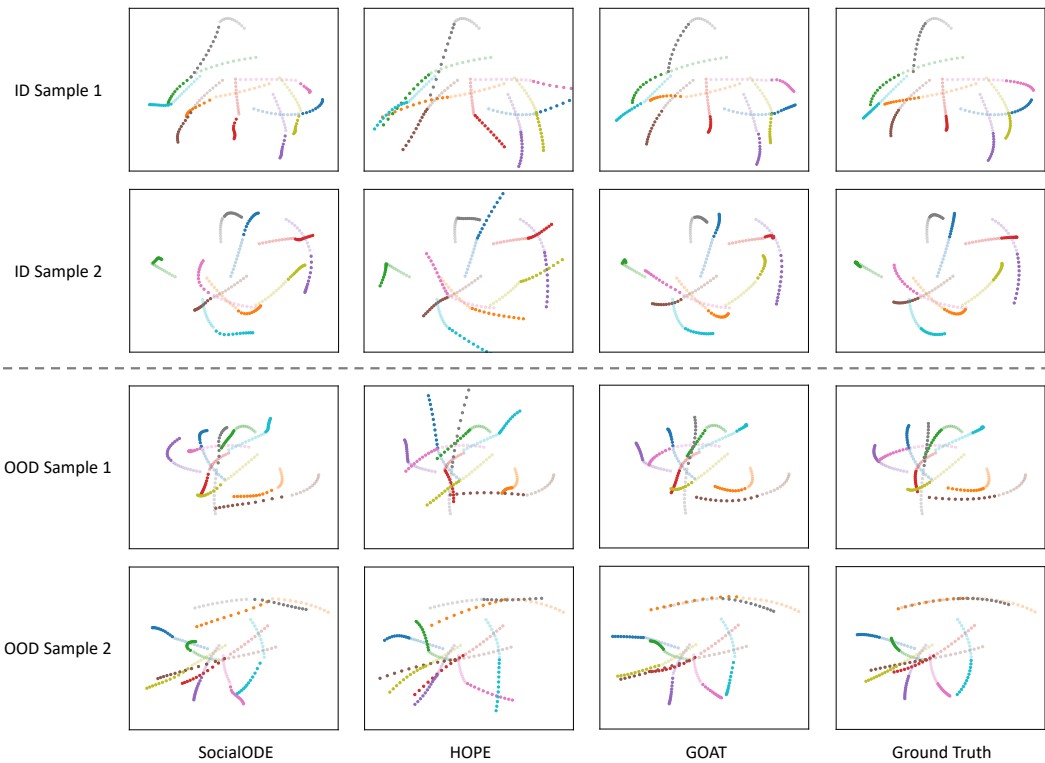

Figure 6: Visualization of different methods on *Springs*. Semi-transparent paths denote observed trajectories, from which the latent initial states are estimated. Solid paths denote model predictions.

