# OpenReview forum: "Graph ODE with Factorized Prototypes for Modeling Complicated Interacting Dynamics"
_ICLR.cc/2024/Conference — Submitted to ICLR 2024_

### Official Review · Reviewer_SJsj · 2023-10-29

**Soundness:** 3 good
**Presentation:** 2 fair
**Contribution:** 2 fair
**Rating:** 6
**Confidence:** 5

**Summary:**

In this paper, the authors propose a variational encoder-decoder model to predict the dynamics of a system of interacting objects given a sequence of historical observations. The system is expressed as a graph, where each node represents an object and the edges represent the interactions between the objects.

The historical trajectories are processed by an attentional GNN encoder to produce the node representations, which are then aggregated to produce a global representation of the system.

A neural ODE decoder is designed to address the problem of modeling the continuous dynamics of interacting systems. The core of the architecture is a graph neural network, in which each node employs a mixture of several globally shared message-passing functions to aggregate information from its neighbors and model the velocity of its latent state. These message-passing functions are called prototypes and are learned from the data. The mixture weights for each node are determined by a function of the node representation and the global representation, which means that the mixture may vary across nodes.

The model is trained using the usual VAE objective. To facilitate the representation learning on the encoder side, the authors propose two auxiliary contrastive losses to encourage the encoder to learn 1) disentangled representations of the objects and the system; and 2) informative system representations. A set of known system parameters is used as a supervisory signal for the second loss.

**Strengths:**

- Nodewise mixture of message-passing functions is an interesting idea and seems novel to me in the context of neural dynamics modeling on graphs.
- The experiments are thorough and convincing. The authors show that the proposed model outperforms the baselines in both in-distribution and out-of-distribution settings. The ablation studies are also helpful in understanding the contributions of the proposed model components. The influence of the hyperparameters is also considered.
- A theoretical analysis about the existence and uniqueness of the solution to the proposed graph ODE is provided.

**Weaknesses:**

- The mixture-of-experts architecture is not well-motivated. While the authors introduce the mixture as a way to enhance the expressivity of the model, it is unclear to me what is the advantage over a single message-passing function that also takes the node representation $\mathbf{u}_i$ and the global representation $\mathbf{g}$ as input.
- GNN-based discrete-time models are not compared in the experiments. It would be interesting to see how the proposed model compares to these models. For example, the NRI model (Kipf et al., 2018) (where the Springs and Charged datasets are from) could be a good baseline.
- Missing related work: https://openreview.net/forum?id=B1lGU64tDr, where GNNs are also used to model the dynamics of interacting objects and similar auxiliary losses are designed for representation learning.

**Questions:**

- The authors claim that the mutual information estimator is trained **adversarially**, but I don't see any adversarial training in Appendix D. The authors should clarify this point.

---

> ### Author Response · Authors · 2023-11-18
> **Response to Reviewer SJsj (I)**
>
> We are truly grateful for the time you have taken to review our paper and your insightful review. Here we address your comments in the following.
>
> > Q1. The mixture-of-experts architecture is not well-motivated. While the authors introduce the mixture as a way to enhance the expressivity of the model, it is unclear to me what is the advantage over a single message-passing function that also takes the node representation and the global representation as input.
>
> A1. Thanks for your comment. We have included a model variant PGODE w. Single, which takes the node representation and the global representation as input with a single message-passing function. The compared performance on the two datasets is recorded as below. From the results, we can observe that the full model performs better than PGODE w. Single, which shows that directly combining representation and the global representation cannot fully capture the complicated dynamics of every node. In contrast, our prototype decomposition can involve different GNN bases, which **model diverse evolving patterns** to jointly determine the individualized dynamics. This strategy can enhance the model expressivity, allowing for more accurate representation learning of hierarchical structures from a mixture-of-experts perspective. We have updated the manuscript accordingly.
>
>
> | Dataset                | Springs (ID) | Springs (ID) | Springs (OOD) | Springs (OOD) | 5AWL (ID) | 5AWL (ID) | 5AWL (ID) | 5AWL (OOD)  | 5AWL (OOD)  | 5AWL (OOD)  |
> |------------------------|------------------|------------------|-------------------|-------------------|------------------|------------------|------------------|-------------------|-------------------|-------------------|
> | Variable     | $q$ | $v$ | $q$ | $v$ | $q_x$ | $q_y$ | $q_z$ | $q_x$ | $q_y$ | $q_z$ |
> | PGODE w. Single | 0.208            | 0.434            | 0.248             | 0.481             | 3.010            | 3.741            | 3.143            | 3.523             | 4.691             | 3.839             |
> | PGODE (Full Model) | **0.070**       | **0.262**       | **0.088**        | **0.291**        | **2.910**       | **3.384**       | **2.904**       | **3.374**        | **4.334**        | **3.615**        |

---

> ### Author Response · Authors · 2023-11-18
> **Response to Reviewer SJsj (II)**
>
> > Q2. GNN-based discrete-time models are not compared in the experiments. It would be interesting to see how the proposed model compares to these models. For example, the NRI model (Kipf et al., 2018) (where the Springs and Charged datasets are from) could be a good baseline.
>
> A2. We have included the baseline AgentFormer [1], NRI [2] and I-GPODE [3] in our performance comparison. The results of these comparisons are presented below and our method outperforms the compared methods, which verifies the superiority of our method. We have updated the manuscript accordingly.
>
>
> | Prediction Length | 12 (ID) | 12 (ID) | 24 (ID) | 24 (ID) | 36 (ID) | 36 (ID) | 12 (OOD) | 12 (OOD) | 24 (OOD) | 24 (OOD) | 36 (OOD) | 36 (OOD) |
> |-------------------|------|------|------|------|------|------|------|------|------|------|------|------|
> | Variable          | $q$  | $v$  | $q$  | $v$  | $q$  | $v$  | $q$  | $v$  | $q$  | $v$  | $q$  | $v$  |
> | _Springs_           |      |      |      |      |      |      |      |      |      |      |      |      |
> | NRI               | 0.103 | 0.425 | 0.210 | 0.681 | 0.693 | 2.263 | 0.119 | 0.472 | 0.246 | 0.770 | 0.807 | 2.406 |
> | AgentFormer       | 0.115 | 0.163 | 0.202 | 0.517 | 1.656 | 1.691 | 0.157 | 0.195 | 0.243 | 0.505 | 1.875 | 1.913 |
> | I-GPODE           | 0.159 | 0.479 | 0.746 | 3.002 | 1.701 | 7.433 | 0.173 | 0.498 | 0.796 | 3.193 | 1.818 | 7.322 |
> | HOPE              | 0.070       | 0.176       | 0.456       | 0.957       | 2.475       | 5.409       | 0.076        | 0.221        | 0.515        | 1.317        | 2.310        | 5.996        |
> | **PGODE (Ours)** | **0.035** | **0.124** | **0.070** | **0.262** | **0.296** | **1.326** | **0.047** | **0.138** | **0.088** | **0.291** | **0.309** | **1.337** |
> | _Charged_           |      |      |      |      |      |      |      |      |      |      |      |      |
> | NRI               | 0.901 | 2.702 | 3.225 | 3.346 | 7.770 | 4.543 | 1.303 | 2.726 | 3.678 | 3.548 | 8.055 | 4.752 |
> | AgentFormer       | 1.076 | 2.476 | 3.631 | 3.044 | 7.513 | 3.944 | 1.384 | 2.514 | 4.224 | 3.199 | 8.985 | 4.002 |
> | I-GPODE           | 1.044 | 2.818 | 3.407 | 3.751 | 7.292 | 4.570 | 1.322 | 2.715 | 3.805 | 3.521 | 8.011 | 4.056 |
> | HOPE              | 0.614       | 2.316       | 3.076       | 3.381       | 8.567       | 8.458       | 0.878        | 2.475        | 3.685        | 3.430        | 10.953       | 9.120        |
> | **PGODE (Ours)** | **0.578** | **2.196** | **2.037** | **2.648** | **4.804** | **3.551** | **0.802** | **2.135** | **2.584** | **2.663** | **5.703** | **3.703** |
>
>
> | Prediction Length | 12 (ID)  | 12 (ID)  | 12 (ID)  | 24 (ID) | 24 (ID)  | 24 (ID)  | 12 (OOD)  | 12 (OOD)  | 12 (OOD)  | 24 (OOD)  | 24 (OOD)  | 24 (OOD)  |
> |-------------------|------|------|------|------|------|------|------|------|------|------|------|------|
> | Variable | $q_x$  | $q_y$  | $q_z$  | $q_x$  | $q_y$  | $q_z$  | $q_x$  | $q_y$  | $q_z$  | $q_x$  | $q_y$  | $q_z$  |
> | _5AWL_    |      |      |      |      |      |      |      |      |      |      |      |      |
> | NRI              | OOM    | OOM    | OOM    | OOM    | OOM    | OOM    | OOM    | OOM    | OOM    | OOM    | OOM    | OOM     |
> | AgentFormer      | OOM    | OOM    | OOM    | OOM    | OOM    | OOM    | OOM    | OOM    | OOM    | OOM    | OOM    | OOM     |
> | I-GPODE          | OOM    | OOM    | OOM    | OOM    | OOM    | OOM    | OOM    | OOM    | OOM    | OOM    | OOM    | OOM     |
> | HOPE             | 2.326  | 2.572  | 2.442  | 3.495  | 3.816  | 3.413  | 2.581  | 3.528  | 2.955  | 4.548  | 5.047  | 4.007   |
> | **PGODE (Ours)** | **2.098** | **2.344** | **2.099** | **2.910** | **3.384** | **2.904** | **2.217** | **3.109** | **2.593** | **3.374** | **4.334** | **3.615**     |
> | _2N5C_    |      |      |      |      |      |      |      |      |      |      |      |      |
> | NRI              | OOM    | OOM    | OOM    | OOM    | OOM    | OOM    | OOM    | OOM    | OOM    | OOM    | OOM    | OOM     |
> | AgentFormer      | OOM    | OOM    | OOM    | OOM    | OOM    | OOM    | OOM    | OOM    | OOM    | OOM    | OOM    | OOM     |
> | I-GPODE          | OOM    | OOM    | OOM    | OOM    | OOM    | OOM    | OOM    | OOM    | OOM    | OOM    | OOM    | OOM     |
> | HOPE             | 1.842  | 1.915  | 2.223  | 2.656  | 2.788  | 3.474  | 2.562  | 2.514  | 2.731  | 3.343  | 3.301  | 3.502  |
> | **PGODE (Ours)** | **1.484** | **1.424** | **1.575** | **1.960** | **2.029** | **2.119** | **1.684** | **1.809** | **1.912** | **2.464** | **2.734** | **2.727**   |

---

> ### Author Response · Authors · 2023-11-18
> **Response to Reviewer SJsj (III)**
>
> > Q3. Missing related work: https://openreview.net/forum?id=B1lGU64tDr, where GNNs are also used to model the dynamics of interacting objects and similar auxiliary losses are designed for representation learning.
>
> A3. Thanks for your comment. We have added the discussion with [4] as follows: "R-SSM [4] models the dynamics of interacting objects using GNNs and includes auxiliary contrastive prediction tasks to enhance discriminative learning while our study focuses on dynamical systems under potential parameter shift and develop a graph ODE framework with high expressivity and generalization capacity."
>
> > Q4. The authors claim that the mutual information estimator is trained adversarially, but I don't see any adversarial training in Appendix D. The authors should clarify this point.
>
> A4. Thanks for your comment. Sorry for the typo. We have revised Appendix D by including: Step 11: Update $\tau'$ in our PGODE using gradient ascent; Step 12: Update other parameters in our PGODE using gradient descent.
>
> **Reference**
>
> [1] Yuan et al., AgentFormer: Agent-Aware Transformers for Socio-Temporal Multi-Agent Forecasting, ICCV 2021.
>
> [2] Kipf et al., Neural Relational Inference for Interacting Systems, ICML 2018.
>
> [3] Yildiz et al., Learning interacting dynamical systems with latent Gaussian process ODEs, NeurIPS 2022.
>
> [4] Yang et al., Relational State-Space Model for Stochastic Multi-Object Systems, ICLR 2020.
>
> In light of these responses, we hope we have addressed your concerns, and hope you will consider raising your score. If there are any additional notable points of concern that we have not yet addressed, please do not hesitate to share them, and we will promptly attend to those points.

---

> > ### Author Response · Authors · 2023-11-22
> > **The deadline for the author-reviewer discussion phase is approaching!**
> >
> > Dear reviewers,
> >
> > We sincerely appreciate your valuable feedback.
> >
> > As the deadline for the author-reviewer discussion phase is approaching, we would like to check if you have any other remaining concerns about our paper. If our responses have adequately addressed your concerns, we kindly hope that you can consider increasing the score.
> >
> > We sincerely thank you for your dedication and effort in evaluating our submission. Please do not hesitate to let us know if you need any clarification or have additional suggestions.
> >
> > Best Regards,
> >
> > Authors.

---

> > > ### Comment · Reviewer_SJsj · 2023-11-22
> > >
> > > Thanks for your clarification and the new experiment results. I have updated my rating to borderline accept based on the revised submission.

---

> > > > ### Author Response · Authors · 2023-11-22
> > > > **Thanks again for your feedback and increasing the rating!**
> > > >
> > > > Thanks again for your feedback and increasing the rating! We are pleased to know that we solve your concerns.
> > > >
> > > > We sincerely thank you for your dedication and effort in evaluating our submission. Please do not hesitate to let us know if you need any clarification or have additional suggestions.

---

### Official Review · Reviewer_RwDt · 2023-10-30

**Soundness:** 3 good
**Presentation:** 1 poor
**Contribution:** 3 good
**Rating:** 5
**Confidence:** 4

**Summary:**

The paper introduces a graph neural net based architecture to model the dynamics of interacting objects. The architecture is an original combination of neural ordinary differential equations (NODEs), graph neural nets (GNNs), and transformer networks.

**Strengths:**

* The way the attentional GNN ideas are incorporated into the Graph NODE framework is novel.

* The way Equation 7 derives system-level features from an aggregation of instance-level features makes intuitive sense.

* The paper conducts experiments  on multiple challenging setups, compares against some state-of-the-art methods and reports strong results.

**Weaknesses:**

* While I appreciate the effort made in the paper to give theoretical guarantees, I have reservations about its correspondence to the mail goal. Lemma 3.1. guarantees only that the ODE has a unique solution, which covers only the computational aspect. This is not a surprising, nor a crucial result. In machine learning approaches to time series modeling, the main challenge usually is not to design ODEs with well-defined solutions. It comes straightforwardly after the Lipschitz continuity assumption that is easy to satisfy when the function approximators are neural nets. The main challenge is rather the predictive power and generalization capacity of the developed models.

 * It is very surprising that despite the extreme similarity of the paper structure, even Figure 1 and Lemma 3.1, to the material reported in (Luo et al., 2023), the paper does not clarify the difference of the proposed method to this recent work. Likewise, the paper aims to solve the exact same problem as Yildiz et al., 2022’s I-GPODE, which even has a subsection about the disentanglement properties of their Graph ODE based interaction model. Yet the proposed method is not differentiated from it. To me even a numerical comparison to such recent work developed for the same exact purpose (interaction modeling) based on the same conjecture (disentanglement) is essential. Overall, I think the literature positioning aspect of the paper is extremely premature.

--- POST REBUTTAL ---

The rebuttal response did not address my concerns. We appear to agree with the authors about the scope of the theoretical contribution of the work but we will remain in disagreement about its significance.

I also do not think my concern about the HONE paper is properly addressed. I still wonder what big weakness of HONE the current approach addresses, where the extreme similarity of the proposed material to this prior work comes, and why it is not addressed properly in the paper. I do not consider the difference in the empirical results as a justification for novelty. I was wondering about the cause, not the effect. I keep my score unchanged.

**Questions:**

* What is the connection between maximum likelihood estimation and representing a distribution by its mean mentioned in Section 3.3 by sentence: **Following the maximum likelihood estimation, here we solely produce …”**

* The ELBO part of the loss makes perfect sense. The way the other two terms are used in Equation 18 are also derived following well-justified desiderata. However, using the three terms together as part of a Bayesian inference objective is another game.  What is the forward model implied by the eventual loss in Equation 18?

* How does the proposed method compare numerically against I-GPODE of (Yildiz et al., 2022)?

Because of a few major weaknesses, I set my initial score below the threshold. However, there is decent potential that these weaknesses can be resolved during the rebuttal, hence my score may increase.

---

> ### Author Response · Authors · 2023-11-18
> **Response to Reviewer RwDt (I)**
>
> We are truly grateful for the time you have taken to review our paper and your insightful review. Here we address your comments in the following.
>
> > Q1. While I appreciate the effort made in the paper to give theoretical guarantees, I have reservations about its correspondence to the main goal. Lemma 3.1. guarantees only that the ODE has a unique solution, which covers only the computational aspect. This is not a surprising, nor a crucial result. In machine learning approaches to time series modeling, the main challenge usually is not to design ODEs with well-defined solutions. It comes straightforwardly after the Lipschitz continuity assumption that is easy to satisfy when the function approximators are neural nets. The main challenge is rather the predictive power and generalization capacity of the developed models.
>
> A1. Thanks for your comment. We agree that the predictive power and generalization capacity are more important, which have been empirically validated by our extensive experiments compared with state-of-the-art approaches. Our theoretical guarantee of well-defined solutions still has merit, which is put forward by the first NODE paper [1] and emphasized by the following works [2]. This theoretical guarantee of the existence and uniqueness would provide insight in designing stable network architecture [2]. However, this is only **a supplementary** to our core contribution, which is the design and application of powerful graph ODEs for modeling dynamics systems, as outlined in Sec. 1 of our paper.

---

> ### Author Response · Authors · 2023-11-18
> **Response to Reviewer RwDt (II)**
>
> > Q2. It is very surprising that despite the extreme similarity of the paper structure, even Figure 1 and Lemma 3.1, to the material reported in (Luo et al., 2023), the paper does not clarify the difference between the proposed method to this recent work. Likewise, the paper aims to solve the exact same problem as Yildiz et al., 2022’s I-GPODE, which even has a subsection about the disentanglement properties of their Graph ODE based interaction model. Yet the proposed method is not differentiated from it. To me even a numerical comparison to such recent work developed for the same exact purpose (interaction modeling) based on the same conjecture (disentanglement) is essential. Overall, I think the literature positioning aspect of the paper is extremely premature.
>
> A2. Thanks for your comment. Firstly, HOPE [3] has been included in the performance comparison. Now we have included a detailed comparison as follows.
> - **Different Objectives**: Our method primarily hones in on modeling dynamical systems in the presence of potential parameter shift. In contrast, HOPE does not encompass this specific challenge.
> - **Different Methodology**: Our method introduces hierarchical context discovery with disentanglement to guide the prototype decomposition of individual nodes in modeling interacting dynamics while HOPE focuses on incorporating second-order graph ODE in evolution modeling.
> - **Different Performance**: From the performance comparison, our proposed method performs much better than HOPE by over 47%.
>
> Secondly, we have included the baselines AgentFormer [4], NRI [5] and I-GPODE [6] in our performance comparison. The results of these comparisons are presented below and our method outperforms the compared methods, which verifies the superiority of our method. We have updated the manuscript accordingly.
>
> | Prediction Length | 12 (ID) | 12 (ID) | 24 (ID) | 24 (ID) | 36 (ID) | 36 (ID) | 12 (OOD) | 12 (OOD) | 24 (OOD) | 24 (OOD) | 36 (OOD) | 36 (OOD) |
> |-------------------|------|------|------|------|------|------|------|------|------|------|------|------|
> | Variable          | $q$  | $v$  | $q$  | $v$  | $q$  | $v$  | $q$  | $v$  | $q$  | $v$  | $q$  | $v$  |
> | _Springs_           |      |      |      |      |      |      |      |      |      |      |      |      |
> | NRI               | 0.103 | 0.425 | 0.210 | 0.681 | 0.693 | 2.263 | 0.119 | 0.472 | 0.246 | 0.770 | 0.807 | 2.406 |
> | AgentFormer       | 0.115 | 0.163 | 0.202 | 0.517 | 1.656 | 1.691 | 0.157 | 0.195 | 0.243 | 0.505 | 1.875 | 1.913 |
> | I-GPODE           | 0.159 | 0.479 | 0.746 | 3.002 | 1.701 | 7.433 | 0.173 | 0.498 | 0.796 | 3.193 | 1.818 | 7.322 |
> | HOPE              | 0.070       | 0.176       | 0.456       | 0.957       | 2.475       | 5.409       | 0.076        | 0.221        | 0.515        | 1.317        | 2.310        | 5.996        |
> | **PGODE (Ours)** | **0.035** | **0.124** | **0.070** | **0.262** | **0.296** | **1.326** | **0.047** | **0.138** | **0.088** | **0.291** | **0.309** | **1.337** |
> | _Charged_           |      |      |      |      |      |      |      |      |      |      |      |      |
> | NRI               | 0.901 | 2.702 | 3.225 | 3.346 | 7.770 | 4.543 | 1.303 | 2.726 | 3.678 | 3.548 | 8.055 | 4.752 |
> | AgentFormer       | 1.076 | 2.476 | 3.631 | 3.044 | 7.513 | 3.944 | 1.384 | 2.514 | 4.224 | 3.199 | 8.985 | 4.002 |
> | I-GPODE           | 1.044 | 2.818 | 3.407 | 3.751 | 7.292 | 4.570 | 1.322 | 2.715 | 3.805 | 3.521 | 8.011 | 4.056 |
> | HOPE              | 0.614       | 2.316       | 3.076       | 3.381       | 8.567       | 8.458       | 0.878        | 2.475        | 3.685        | 3.430        | 10.953       | 9.120        |
> | **PGODE (Ours)** | **0.578** | **2.196** | **2.037** | **2.648** | **4.804** | **3.551** | **0.802** | **2.135** | **2.584** | **2.663** | **5.703** | **3.703** |

---

> ### Author Response · Authors · 2023-11-18
> **Response to Reviewer RwDt (III)**
>
> | Prediction Length | 12 (ID)  | 12 (ID)  | 12 (ID)  | 24 (ID) | 24 (ID)  | 24 (ID)  | 12 (OOD)  | 12 (OOD)  | 12 (OOD)  | 24 (OOD)  | 24 (OOD)  | 24 (OOD)  |
> |-------------------|------|------|------|------|------|------|------|------|------|------|------|------|
> | Variable | $q_x$  | $q_y$  | $q_z$  | $q_x$  | $q_y$  | $q_z$  | $q_x$  | $q_y$  | $q_z$  | $q_x$  | $q_y$  | $q_z$  |
> | _5AWL_    |      |      |      |      |      |      |      |      |      |      |      |      |
> | NRI              | OOM    | OOM    | OOM    | OOM    | OOM    | OOM    | OOM    | OOM    | OOM    | OOM    | OOM    | OOM     |
> | AgentFormer      | OOM    | OOM    | OOM    | OOM    | OOM    | OOM    | OOM    | OOM    | OOM    | OOM    | OOM    | OOM     |
> | I-GPODE          | OOM    | OOM    | OOM    | OOM    | OOM    | OOM    | OOM    | OOM    | OOM    | OOM    | OOM    | OOM     |
> | HOPE             | 2.326  | 2.572  | 2.442  | 3.495  | 3.816  | 3.413  | 2.581  | 3.528  | 2.955  | 4.548  | 5.047  | 4.007   |
> | **PGODE (Ours)** | **2.098** | **2.344** | **2.099** | **2.910** | **3.384** | **2.904** | **2.217** | **3.109** | **2.593** | **3.374** | **4.334** | **3.615**     |
> | _2N5C_    |      |      |      |      |      |      |      |      |      |      |      |      |
> | NRI              | OOM    | OOM    | OOM    | OOM    | OOM    | OOM    | OOM    | OOM    | OOM    | OOM    | OOM    | OOM     |
> | AgentFormer      | OOM    | OOM    | OOM    | OOM    | OOM    | OOM    | OOM    | OOM    | OOM    | OOM    | OOM    | OOM     |
> | I-GPODE          | OOM    | OOM    | OOM    | OOM    | OOM    | OOM    | OOM    | OOM    | OOM    | OOM    | OOM    | OOM     |
> | HOPE             | 1.842  | 1.915  | 2.223  | 2.656  | 2.788  | 3.474  | 2.562  | 2.514  | 2.731  | 3.343  | 3.301  | 3.502  |
> | **PGODE (Ours)** | **1.484** | **1.424** | **1.575** | **1.960** | **2.029** | **2.119** | **1.684** | **1.809** | **1.912** | **2.464** | **2.734** | **2.727**   |
>
> Lastly, we have included more recent literature in our related works as:
> "AgentFormer [4] jointly models both time and social dimensions with semantic information preserved. NRI [5] models interactions along with node states from observations using GNNs. I-GPODE [6] estimates the uncertainty of trajectory predictions using the Gaussian process, which facilitates effective long-term predictions. R-SSM [7] models the dynamics of interacting objects using GNNs and includes auxiliary contrastive prediction tasks to enhance discriminative learning."
>
> > Q3. What is the connection between maximum likelihood estimation and representing a distribution by its mean mentioned in Section 3.3 by sentence: Following the maximum likelihood estimation, here we solely produce …”
>
> A3. Thanks for your comment. Here, we assume that the distribution of
> $\hat{x}^t_i$  follows an Gaussian distribution with variance $\sigma^2$ and maximizing the likelihood estimation can get
> $\min$ $\sum_r$ $\frac{ ||x^t_i-\hat{x}^t_i (r) ||^{2}}{2 \sigma^{2}}$, where $\hat{x}^t_i (r)$ denotes the r-th sample from the distribution. Minimizing this is equivalent to $\min \frac{||x^{t}_i-\mu^{t}_i||^{2}}{2 \sigma^{2}}$, where $\mu^{t}_i$ denotes the mean of $\hat{x}^t_i (r)$.
>
> > Q4. The ELBO part of the loss makes perfect sense. The way the other two terms are used in Equation 18 is also derived following well-justified desiderata. However, using the three terms together as part of a Bayesian inference objective is another game. What is the forward model implied by the eventual loss in Equation 18?
>
> A4. Thanks for your comment. In Eqn. 18, the other two loss terms serve as a regularization mechanism using mutual information to constrain the model parameters. This strategy is popular in recent Bayesian inference models [8,9]. Therefore, our integration improves the predictive performance while maintaining a robust Bayesian inference framework.
>
> > Q5. How does the proposed method compare numerically against I-GPODE of (Yildiz et al., 2022)?
>
> A5. Thanks for your comment. We have included the baseline I-GPODE in A1.

---

> ### Author Response · Authors · 2023-11-18
> **Response to Reviewer RwDt (IV)**
>
> **Reference**
>
> [1] Ricky T. Q. Chen, Neural Ordinary Differential Equations, NeurIPS 2018.
>
> [2] Kong et al., Sde-net: Equipping deep neural networks with uncertainty estimates, ICML 2020.
>
> [3] Luo et al., Hope: High-order graph ode for modeling interacting dynamics. ICML 2023.
>
> [4] Yuan et al., AgentFormer: Agent-Aware Transformers for Socio-Temporal Multi-Agent Forecasting, ICCV 2021.
>
> [5] Kipf et al., Neural Relational Inference for Interacting Systems, ICML 2018.
>
> [6] Yildiz et al., Learning interacting dynamical systems with latent Gaussian process ODEs, NeurIPS 2022.
>
> [7] Yang et al., Relational State-Space Model for Stochastic Multi-Object Systems, ICLR 2020.
>
> [8] Rhodes et al., Local Disentanglement in Variational Auto-Encoders Using Jacobian L1 Regularization, NeurIPS 2021.
>
> [9] Xu et al., Multi-objects Generation with Amortized Structural Regularization, NeurIPS 2019.
>
> In light of these responses, we hope we have addressed your concerns, and hope you will consider raising your score. If there are any additional notable points of concern that we have not yet addressed, please do not hesitate to share them, and we will promptly attend to those points.

---

> > ### Author Response · Authors · 2023-11-22
> > **The deadline for the author-reviewer discussion phase is approaching!**
> >
> > Dear reviewers,
> >
> > We sincerely appreciate your valuable feedback.
> >
> > As the deadline for the author-reviewer discussion phase is approaching, we would like to check if you have any other remaining concerns about our paper. If our responses have adequately addressed your concerns, we kindly hope that you can consider increasing the score.
> >
> > We sincerely thank you for your dedication and effort in evaluating our submission. Please do not hesitate to let us know if you need any clarification or have additional suggestions.
> >
> > Best Regards,
> >
> > Authors.

---

> ### Author Response · Authors · 2023-11-23
> **Thank you for your invaluable feedback!**
>
> Dear Reviewer,
>
>
> Thank you for your invaluable feedback. As the deadline for the author-reviewer discussion phase is approaching, we hope to make sure that our response sufficiently addressed your concerns regarding the literature positioning, as well as the revised version of our paper. We hope this could align with your expectations and positively influence the score. Please do not hesitate to let us know if you need any clarification or have additional suggestions.
>
>
> Best Regards,
>
> Authors

---

### Official Review · Reviewer_ANXm · 2023-11-04

**Soundness:** 3 good
**Presentation:** 3 good
**Contribution:** 3 good
**Rating:** 6
**Confidence:** 4

**Summary:**

This paper delves into the modeling of interacting dynamical systems, employing Graph Neural Networks (GNNs) as a fundamental tool. A novel approach, termed GOAT, is introduced, which capitalizes on disentangled contexts to formulate factorized prototypes for graph ODEs, aiming for heightened expressivity and improved generalization. GOAT meticulously extracts both object-level and system-level contexts through an attention-based GNN framework. The incorporation of disentangled representation learning alongside a mixture of experts strategy further propels the system's generalization and expressivity. Through rigorous experimentation on physical and molecular dynamical systems, the paper substantiates that GOAT surpasses existing methods in performance.

**Strengths:**

The technical robustness of the proposed method is commendable. Additionally, the paper goes the extra mile by conducting an exhaustive ablation study, ensuring a thorough validation of each model component's effectiveness, which adds a layer of credibility to the findings.

**Weaknesses:**

On the flip side, the complexity of the proposed method is rather high. The integration of a mixture of experts not only elevates the intricacy but also augments the number of parameters, consequently escalating the training cost when juxtaposed with the baseline methods. A comparative analysis with an ensemble of baseline methods might present a fairer landscape for evaluation, given the escalated complexity and training cost.

**Questions:**

The training cost is a pivotal factor for practical implementation. How does the training cost of the proposed method compare with that of the baseline methods? This comparison would provide a clearer understanding of the practical implications entailed by the proposed method.

---

> ### Author Response · Authors · 2023-11-18
> **Response to Reviewer ANXm**
>
> We are truly grateful for the time you have taken to review our paper, your insightful comments and support. Your positive feedback is incredibly encouraging for us! In the following response, we would like to address your major concern and provide additional clarification.
>
> > Q1. On the flip side, the complexity of the proposed method is rather high. The integration of a mixture of experts not only elevates the intricacy but also augments the number of parameters, consequently escalating the training cost when juxtaposed with the baseline methods. A comparative analysis with an ensemble of baseline methods might present a fairer landscape for evaluation, given the escalated complexity and training cost.
>
> A1. Thanks for your comment. We have included the comparison of computation cost as follows and we can observe that our method has a competitive computation cost. In particular, the performance of HOPE is much worse than ours (the performance increasement of ours is over 47% compared with HOPE), while our computational burden only increases a little. Moreover, both the performance and efficiency of I-GPODE are worse than ours.
>
> | Method  | LSTM | GRU  | NODE | LG-ODE | MPNODE | SocialODE | I-GPODE | HOPE  | PGODE (Ours) |
> | ------- | ---- | ---- | ---- | ------ | ------ | --------- | ------- | ----- | ----------- |
> | Springs | 1.53 | 1.04 | 2.21 | 17.39  | 23.33  | 21.02     | 267.08  | 23.86 | 37.03       |
> | Charged | 1.33 | 1.02 | 2.06 | 16.59  |  22.26 | 19.93     | 250.23  | 20.43 | 33.88       |
>
> > Q2. The training cost is a pivotal factor for practical implementation. How does the training cost of the proposed method compare with that of the baseline methods? This comparison would provide a clearer understanding of the practical implications entailed by the proposed method.
>
> A2. Thanks for your question. We have included the comparison in A1.
>
> Thanks again for appreciating our work and for your constructive suggestions. Please let us know if you have further questions.

---

> > ### Author Response · Authors · 2023-11-22
> > **The deadline for the author-reviewer discussion phase is approaching!**
> >
> > Dear reviewers,
> >
> > We sincerely appreciate your valuable feedback.
> >
> > As the deadline for the author-reviewer discussion phase is approaching, we would like to check if you have any other remaining concerns about our paper. If our responses have adequately addressed your concerns, we kindly hope that you can consider increasing the score.
> >
> > We sincerely thank you for your dedication and effort in evaluating our submission. Please do not hesitate to let us know if you need any clarification or have additional suggestions.
> >
> > Best Regards,
> >
> > Authors.

---

> ### Author Response · Authors · 2023-11-23
> **Thank you for your invaluable feedback!**
>
> Dear Reviewer,
>
> Thank you for your invaluable feedback. As the deadline for the author-reviewer discussion phase is approaching, we hope to make sure that our response sufficiently addressed your concerns regarding the complexity, as well as the revised version of our paper. We hope this could align with your expectations and positively influence the score. Please do not hesitate to let us know if you need any clarification or have additional suggestions.
>
> Best Regards,
>
> Authors

---

### Official Review · Reviewer_bPDi · 2023-11-05

**Soundness:** 2 fair
**Presentation:** 1 poor
**Contribution:** 2 fair
**Rating:** 3
**Confidence:** 2

**Summary:**

This paper proposes a model for interacting dynamical systems. Their approach uses a Graph ODE with factorized prototypes and disentangled system and object level representations to enable greater generalization.

**Strengths:**

The paper outlines the key features needed in a system for predicting interacting dynamics
1) Capturing continuous dynamics
2) Being expressive enough to capture complex dynamics
3) Generalizing out of distribution.
They introduce different GNNs for different interacting prototypes with different updating rules.

**Weaknesses:**

This paper is very hard to read and uses jargon that most ML people may not be familiar with. I believe this paper is trying to do not just interacting dynamics but actually agent-based modeling. Given this context, I am completely unclear as to what a "factorized prototype" is, the authors just use this term without definition and repeatedly explain the method this way. The other key feature of an agent based system thats not just in an interactive dynamic system (like a gene regulatory network) is that the graph consisting of agents changes with time based on the movements of the agents. This would be the key challenge to using a GNN/Graph ODE to model agent dynamics and this is not  addressed here clearly despite defining the data as consisting of different graphs at different timepoints.

Additionally the authors refer to some type of hierarchical representation which is again completely unclear. My best guess is that each object somehow evolves based on a combination of those GNN "factored prototypes" but that seems somewhat strange. Couldn't we just learn an individualized evolution based on a GNN with an MLP aggregation layer? What is the need for "factored" prototypes, I can at least see an argument for selecting a prototype (using a softmax or something of this sort).

The problem setup seems to sweep the most important thing under the rug. They talk about the problem of forecasting the dynamics of predicting the features, but what about continuously predicting the changes in the graph? Or are these all given? Aren't their observations discrete? Don't the graphs have to be updated continuously also, and how exactly is the graph changed at each iteration of the graph ODE? These and many question are unanswered.

I would recommend for the authors to rewrite the paper from scratch, not assuming anyone knows anything about agent dynamics or object contexts. Define every technical term, and fully specify the problem. Without a thorough rewrite it is very unclear what is going on.

Moreoever there are other agent based models like the agent former that the authors have not compared their method with.

**Questions:**

Exactly what sort of dynamics are you thinking of modeling and how does the graph changing over time get modelled in your system?

---

> ### Author Response · Authors · 2023-11-18
> **Response to Reviewer bPDi (I)**
>
> We are truly grateful for the time you have taken to review our paper and your insightful review. Here we address your comments in the following.
>
> > Q1. This paper is very hard to read and uses jargon that most ML people may not be familiar with. I believe this paper is trying to do not just interacting dynamics but actually agent-based modeling. Given this context, I am completely unclear as to what a "factorized prototype" is, the authors just use this term without definition and repeatedly explain the method this way. The other key feature of an agent based system that's not just in an interactive dynamic system (like a gene regulatory network) is that the graph consisting of agents changes with time based on the movements of the agents. This would be the key challenge to using a GNN/Graph ODE to model agent dynamics and this is not addressed here clearly despite defining the data as consisting of different graphs at different timepoints.
>
> A1. Thanks for your comment. *Firstly*, let me clarify "factorized prototype", which refers to the decomposition of an agent's behavior within a system into several fundamental elements. In our context, each GNN within the graph ODE framework acts as a base model (a.k.a., prototype). The interaction dynamics of each agent are then represented as a weighted combination of these prototypes, which is formulated as our ODE function (Eqn. 10). This concept of "factorized prototype" has been utilized in several previous works [1,2]. However, to improve clarity, we have revised this terminology to "**prototype decomposition**," a more widely used term in the field of machine learning. We also rename our method as Prototypical Graph ODE (PGODE). *Secondly*, we agree that the evolution of graphs over time is a significant challenge in dynamical system modeling. However, our current research centers on **predicting the trajectories** from graph-structured data rather than inferring dynamical graphs, which is aligned with [3,4,5,6,7]. In particular, in Springs and Charged, the fixed graphs are given based on real springs or electric charge effects rather than node positions. We acknowledge the importance of graph inference in more complicated scenarios and plan to explore it in future work, which has been stated in the revised version of our paper.
>
>
>
> > Q2. Additionally the authors refer to some type of hierarchical representation which is again completely unclear. My best guess is that each object somehow evolves based on a combination of those GNN "factored prototypes" but that seems somewhat strange. Couldn't we just learn an individualized evolution based on a GNN with an MLP aggregation layer? What is the need for "factored" prototypes, I can at least see an argument for selecting a prototype (using a softmax or something of this sort).
>
>
> A2. Thanks for your comment. To address your concern, we have included a model variant PGODE w. MLP, which combines a GNN with an MLP to learn the individualized dynamics. The compared performance on the two datasets is recorded as below. From the results, we can observe that the full model performs better than PGODE w. MLP, which shows that a straightforward MLP cannot fully capture the complicated dynamics of every node. In contrast, our prototype decomposition can involve different GNN bases, which **model diverse evolving patterns** to jointly determine the individualized dynamics. This strategy can enhance the model expressivity, allowing for more accurate representation learning of hierarchical structures from a mixture-of-experts perspective. We have updated the manuscript accordingly.
>
> | Dataset                | Springs (ID) | Springs (ID) | Springs (OOD) | Springs (OOD) | 5AWL (ID) | 5AWL (ID) | 5AWL (ID) | 5AWL (OOD)  | 5AWL (OOD)  | 5AWL (OOD)  |
> |------------------------|------------------|------------------|-------------------|-------------------|------------------|------------------|------------------|-------------------|-------------------|-------------------|
> | Variable     | $q$ | $v$ | $q$ | $v$ | $q_x$ | $q_y$ | $q_z$ | $q_x$ | $q_y$ | $q_z$ |
> | PGODE w. MLP    | 0.152            | 0.454            | 0.179             | 0.514             | 2.997            | 3.638            | 3.240            | 3.605             | 4.492             | 3.908             |
> | PGODE (Full Model) | **0.070**       | **0.262**       | **0.088**        | **0.291**        | **2.910**       | **3.384**       | **2.904**       | **3.374**        | **4.334**        | **3.615**        |

---

> ### Author Response · Authors · 2023-11-18
> **Response to Reviewer bPDi (II)**
>
> > Q3. The problem setup seems to sweep the most important thing under the rug. They talk about the problem of forecasting the dynamics of predicting the features, but what about continuously predicting the changes in the graph? Or are these all given? Aren't their observations discrete? Don't the graphs have to be updated continuously also, and how exactly is the graph changed at each iteration of the graph ODE? These and many questions are unanswered.
>
> A3. Thanks for your comment. Here, although the observations are discrete, the underlying dynamics should be still continuous. Modeling continuous trajectories enables us to extrapolate and predict outputs at any given point in time, effectively **bridging the gap between discrete observations and continuous underlying dynamics**. As for **graphs** in physical simulations, the graph structures remain fixed, which are from **real springs or electric charge effects** following previous works [3,7]. In molecular simulations, the graph structures remain stable for a small period in reality, and therefore, we update the graph structure based on node position every $T$ timestamps. Modeling interacting dynamics with fixed or robust graphs is an important task, which has received extensive attention as in [3,4,5,6,7]. We have updated the manuscript accordingly.
>
>
> > Q4. I would recommend for the authors to rewrite the paper from scratch, not assuming anyone knows anything about agent dynamics or object contexts. Define every technical term, and fully specify the problem. Without a thorough rewrite it is very unclear what is going on.
>
> A4. Thanks for your comment. We have rewritten the abstract, introduction and problem definition to make sure everything is clear. In particular, we have mainly rewritten the following parts.
>
> - **Abstract.** This paper studies the problem of modeling multi-agent dynamical systems, where agents could interact mutually to influence their behaviors. Recent research predominantly uses geometric graphs to depict these mutual interactions, which are then captured by powerful graph neural networks (GNNs). ... The core of PGODE is to incorporate prototype decomposition into a continuous graph ODE framework, which is guided by contextual knowledge from historical trajectories.
> - **Introduction (First paragraph)** Multi-agent dynamical systems are ubiquitous in the real world where agents can be vehicles and microcosmic particles. These agents could have complicated interactions resulting from behavioral or mechanical influences, which result in complicated future trajectories of the whole system. Modeling the interacting dynamics is a crucial challenge in machine learning with broad applications in fluid mechanics, autonomous driving and human-robot interactions.
> - **Introduction (Second paragraph)** The core of PGODE lies in exploring disentangled contexts, i.e., object states and system states, inferred from historical trajectories for graph ODE with high expressivity and generalization.
> - **Problem Definitation**. Given a multi-agent dynamical system, we characterize the agent states and interaction at the $t$-th timestamp as a graph $G^t = (\mathcal{V}, \mathcal{E}^t, \textbf{X}^t)$, where each node in $\mathcal{V}$ is an object, $\mathcal{E}^t$ comprises all the edges and $\textbf{X}^t$ is the object attribute matrix. $N$ represents the number of objects. Given the observations $G^{1:T_{obs}} = \{G^1, \cdots, G^{T_{obs}} \}$, our goal is to learn a model capable of predicting the future trajectories $\textbf{X}^{T_{obs}+1: T}$. Our interacting dynamics system is governed by a set of equations with system parameters denoted as $\xi$. Different values of $\xi$ could influence underlying dynamical principles, leading to potential shift in trajectory distributions. Therefore, it is essential to extract contextual information related to both system parameters and node states from historical observations for faithful trajectory predictions.

---

> ### Author Response · Authors · 2023-11-18
> **Response to Reviewer bPDi (III)**
>
> > Q5. Moreover, there are other agent based models like the agent former that the authors have not compared their method with.
>
> A5. Thanks for your comment. We have included the baseline NRI [7], AgentFormer [8], and I-GPODE [9] in our performance comparison. The results of these comparisons are presented below and our method outperforms the compared methods, which verifies the superiority of our method. We have updated the manuscript accordingly.
>
>
> | Prediction Length | 12 (ID) | 12 (ID) | 24 (ID) | 24 (ID) | 36 (ID) | 36 (ID) | 12 (OOD) | 12 (OOD) | 24 (OOD) | 24 (OOD) | 36 (OOD) | 36 (OOD) |
> |-------------------|------|------|------|------|------|------|------|------|------|------|------|------|
> | Variable         |  $q$  | $v$  | $q$  | $v$  | $q$  | $v$  | $q$  | $v$  | $q$  | $v$  | $q$  | $v$  |
> | _Springs_           |      |      |      |      |      |      |      |      |      |      |      |      |
> | NRI               | 0.103 | 0.425 | 0.210 | 0.681 | 0.693 | 2.263 | 0.119 | 0.472 | 0.246 | 0.770 | 0.807 | 2.406 |
> | AgentFormer       | 0.115 | 0.163 | 0.202 | 0.517 | 1.656 | 1.691 | 0.157 | 0.195 | 0.243 | 0.505 | 1.875 | 1.913 |
> | I-GPODE           | 0.159 | 0.479 | 0.746 | 3.002 | 1.701 | 7.433 | 0.173 | 0.498 | 0.796 | 3.193 | 1.818 | 7.322 |
> | HOPE              | 0.070       | 0.176       | 0.456       | 0.957       | 2.475       | 5.409       | 0.076        | 0.221        | 0.515        | 1.317        | 2.310        | 5.996        |
> | **PGODE (Ours)** | **0.035** | **0.124** | **0.070** | **0.262** | **0.296** | **1.326** | **0.047** | **0.138** | **0.088** | **0.291** | **0.309** | **1.337** |
> | _Charged_           |      |      |      |      |      |      |      |      |      |      |      |      |
> | NRI               | 0.901 | 2.702 | 3.225 | 3.346 | 7.770 | 4.543 | 1.303 | 2.726 | 3.678 | 3.548 | 8.055 | 4.752 |
> | AgentFormer       | 1.076 | 2.476 | 3.631 | 3.044 | 7.513 | 3.944 | 1.384 | 2.514 | 4.224 | 3.199 | 8.985 | 4.002 |
> | I-GPODE           | 1.044 | 2.818 | 3.407 | 3.751 | 7.292 | 4.570 | 1.322 | 2.715 | 3.805 | 3.521 | 8.011 | 4.056 |
> | HOPE              | 0.614       | 2.316       | 3.076       | 3.381       | 8.567       | 8.458       | 0.878        | 2.475        | 3.685        | 3.430        | 10.953       | 9.120        |
> | **PGODE (Ours)** | **0.578** | **2.196** | **2.037** | **2.648** | **4.804** | **3.551** | **0.802** | **2.135** | **2.584** | **2.663** | **5.703** | **3.703** |
>
>
> | Prediction Length | 12 (ID)  | 12 (ID)  | 12 (ID)  | 24 (ID) | 24 (ID)  | 24 (ID)  | 12 (OOD)  | 12 (OOD)  | 12 (OOD)  | 24 (OOD)  | 24 (OOD)  | 24 (OOD)  |
> |-------------------|------|------|------|------|------|------|------|------|------|------|------|------|
> | Variable | $q_x$  | $q_y$  | $q_z$  | $q_x$  | $q_y$  | $q_z$  | $q_x$  | $q_y$  | $q_z$  | $q_x$  | $q_y$  | $q_z$  |
> | _5AWL_    |      |      |      |      |      |      |      |      |      |      |      |      |
> | NRI              | OOM    | OOM    | OOM    | OOM    | OOM    | OOM    | OOM    | OOM    | OOM    | OOM    | OOM    | OOM     |
> | AgentFormer      | OOM    | OOM    | OOM    | OOM    | OOM    | OOM    | OOM    | OOM    | OOM    | OOM    | OOM    | OOM     |
> | I-GPODE          | OOM    | OOM    | OOM    | OOM    | OOM    | OOM    | OOM    | OOM    | OOM    | OOM    | OOM    | OOM     |
> | HOPE             | 2.326  | 2.572  | 2.442  | 3.495  | 3.816  | 3.413  | 2.581  | 3.528  | 2.955  | 4.548  | 5.047  | 4.007   |
> | **PGODE (Ours)** | **2.098** | **2.344** | **2.099** | **2.910** | **3.384** | **2.904** | **2.217** | **3.109** | **2.593** | **3.374** | **4.334** | **3.615**     |
> | _2N5C_    |      |      |      |      |      |      |      |      |      |      |      |      |
> | NRI              | OOM    | OOM    | OOM    | OOM    | OOM    | OOM    | OOM    | OOM    | OOM    | OOM    | OOM    | OOM     |
> | AgentFormer      | OOM    | OOM    | OOM    | OOM    | OOM    | OOM    | OOM    | OOM    | OOM    | OOM    | OOM    | OOM     |
> | I-GPODE          | OOM    | OOM    | OOM    | OOM    | OOM    | OOM    | OOM    | OOM    | OOM    | OOM    | OOM    | OOM     |
> | HOPE             | 1.842  | 1.915  | 2.223  | 2.656  | 2.788  | 3.474  | 2.562  | 2.514  | 2.731  | 3.343  | 3.301  | 3.502  |
> | **PGODE (Ours)** | **1.484** | **1.424** | **1.575** | **1.960** | **2.029** | **2.119** | **1.684** | **1.809** | **1.912** | **2.464** | **2.734** | **2.727**   |

---

> ### Author Response · Authors · 2023-11-18
> **Response to Reviewer bPDi (IV)**
>
> > Q6. Exactly what sort of dynamics are you thinking of modeling and how does the graph changing over time get modelled in your system?
>
> A6. Thanks for your question. Modeling interacting dynamics with fixed or robust graphs is an important task, which has received extensive attention as in [3,4,5,6,7]. Here, in physical datasets, we model the dynamics of objects interacted by **connected springs or electric charge effects** for trajectory predictions. Therefore, the graph is given and fixed. In molecular datasets, we model the dynamics of atoms and their interactions are related to the pairwise distance. In reality, the graph structures remain stable for a small period and we update the graph structure based on node position every $T$ timestamps. We have updated the manuscript accordingly.
>
>
>
>
>
> **Reference**
>
> [1] Das S, et al., Interpreting deep neural networks through prototype factorization, ICDM 2020.
>
> [2] Zhao J, et al. ProtoViewer: Visual interpretation and diagnostics of deep neural networks with factorized prototypes, IEEE Visualization Conference 2020.
>
> [3] Huang et al, Learning Continuous System Dynamics from Irregularly-Sampled Partial Observations, NeurIPS 2020.
>
> [4] Fang et al., Spatial-Temporal Graph ODE Networks for Traffic Flow Forecasting, KDD 2021.
>
> [5] Cao et al., Efficient Learning of Mesh-Based Physical Simulation with Bi-Stride Multi-Scale Graph Neural Network, ICML 2023
>
> [6] Han et al., Predicting Physics in Mesh-reduced Space with Temporal Attention, ICLR 2022
>
> [7] Kipf et al., Neural Relational Inference for Interacting Systems, ICML 2018.
>
> [8] Yuan et al., AgentFormer: Agent-Aware Transformers for Socio-Temporal Multi-Agent Forecasting, ICCV 2021.
>
> [9] Yildiz et al., Learning interacting dynamical systems with latent Gaussian process ODEs, NeurIPS 2022.
>
> In light of these responses, we hope we have addressed your concerns, and hope you will consider raising your score. If there are any additional notable points of concern that we have not yet addressed, please do not hesitate to share them, and we will promptly attend to those points.

---

> > ### Author Response · Authors · 2023-11-22
> > **The deadline for the author-reviewer discussion phase is approaching!**
> >
> > Dear reviewers,
> >
> > We sincerely appreciate your valuable feedback.
> >
> > As the deadline for the author-reviewer discussion phase is approaching, we would like to check if you have any other remaining concerns about our paper. If our responses have adequately addressed your concerns, we kindly hope that you can consider increasing the score.
> >
> > We sincerely thank you for your dedication and effort in evaluating our submission. Please do not hesitate to let us know if you need any clarification or have additional suggestions.
> >
> > Best Regards,
> >
> > Authors.

---

> ### Author Response · Authors · 2023-11-23
> **Thank you for your invaluable feedback!**
>
> Dear Reviewer,
>
>
> Thank you for your invaluable feedback. As the deadline for the author-reviewer discussion phase is approaching, we hope to make sure that our response sufficiently addressed your concerns regarding the presentation, as well as the revised version of our paper. We hope this could align with your expectations and positively influence the score. Please do not hesitate to let us know if you need any clarification or have additional suggestions.
>
>
> Best Regards,
>
> Authors

---

### Author Response · Authors · 2023-11-19
**General Response**

Dear Reviewers,

We thank you for your careful reviews and constructive suggestions. We acknowledge the positive comments such as "**outlines the key features**" (Reviewer bPDi), “**technical robustness**” (Reviewer ANXm), "**a thorough validation and credibility**" (Reviewer ANXm), "**novel idea**” (Reviewer RwDt), "**makes intuitive sense**” (Reviewer RwDt), "**strong results**” (Reviewer RwDt), "**interesting idea**” (Reviewer SJsj), “**experiments are thorough and convincing**” (Reviewer SJsj) and “**theoretical analysis**” (Reviewer SJsj).

We have made extensive revisions based on these valuable comments, which we briefly summarize below for your convenience, and have accordingly revised our article with major changes highlighted in **blue**.

- We have **clarified our problem settings, introduced concepts and methodology details** by rewriting the introduction, abstract, problem definition and method to make the paper more clear.

- We have added **more ablation studies** including combining a GNN with an MLP and representation combination to show the effectiveness of our proposed component.

- We have included **more competing baselines** including AgentForme, NRI and I-GPODE to demonstrate the superiority of our approach.

- We have included **more related works** as reviewers suggested and made extensive comparison in our discussion to make the paper complete.

- We have included **efficiency comparison** between our method and various baselines and observed that our method shows competitive efficiency.

- We have **proofread** the manuscript to correct some typos and mistakes.

We have also responded to your questions point by point. Once again, we greatly appreciate your effort and time spent revising our manuscript. Please let us know if you have any follow-up questions. We will be happy to answer them.

Best regards,

the Authors

---

> ### Author Response · Authors · 2023-11-23
> **The deadline for the author-reviewer discussion phase is approaching!**
>
> Dear reviewers,
>
> We sincerely appreciate your valuable feedback.
>
> As the deadline for the author-reviewer discussion phase is approaching, we would like to check if you have any other remaining concerns about our paper.
>
> We sincerely thank you for your dedication and effort in evaluating our submission. Please do not hesitate to let us know if you need any clarification or have additional suggestions.
>
> Best Regards,
>
> Authors.

---

### Comment · Senior_Area_Chairs · 2023-11-23
**author-reviewer discussion about to end**

Dear reviewers, as the deadline for the author-reviewer discussion phase is approaching, please take a look at the authors' reply (if you have not done so) and see if your concerns have been addressed.

---

### Meta-Review · Area_Chair_KKhk · 2023-12-10

**Metareview:**

This paper proposes a new approach called Graph ODE with factorized prototypes (GOAT) to model interacting dynamical systems, which is critical for understanding physical dynamics and biological processes. GOAT incorporates factorized prototypes from contextual knowledge into a continuous graph ODE framework, which allows for explicit modeling the independent influence of object-level and system-level contexts and enhances generalization capability under system changes. The model is optimized using an end-to-end variational inference framework and is shown to outperform existing methods in both in-distribution and out-of-distribution settings.

Firstly, the studied problem is an important one. Secondly, the paper contains some novel ideas, such as system-level features in equation 7, and also provides some theoretical analysis. In addition, experiments are conducted on several different setups, which shows the GOAT outperforms several baselines.

Firstly, the novelty of this paper is limited, since it contains extremely similar materials to existing works. Secondly, the contribution of the theoretical guarantees is not significant, since the main concern is the generalization capacity of the model. Finally, the paper should be further polished more carefully, such as providing standard deviations of experiments for checking statistical significance.

**Justification For Why Not Higher Score:**

The novelty of this paper is limited, and the contribution of the theoretical guarantees is not significant.

**Justification For Why Not Lower Score:**

N/A

---

### Decision · Program_Chairs · 2024-01-16

Reject